# Human Expertise in Algorithmic Prediction

**Rohan Alur**
EECS, LIDS
MIT
ralur@mit.edu

**Manish Raghavan**
EECS, LIDS, Sloan
MIT
mragh@mit.edu

**Devavrat Shah**
EECS, IDSS, LIDS, SDSC
MIT
devavrat@mit.edu

## Abstract

We introduce a novel framework for incorporating human expertise into algorithmic predictions. Our approach leverages human judgment to distinguish inputs which are *algorithmically indistinguishable*, or "look the same" to predictive algorithms. We argue that this framing clarifies the problem of human-AI collaboration in prediction tasks, as experts often form judgments by drawing on information which is not encoded in an algorithm's training data. Algorithmic indistinguishability yields a natural test for assessing whether experts incorporate this kind of "side information", and further provides a simple but principled method for selectively incorporating human feedback into algorithmic predictions. We show that this method provably improves the performance of any feasible algorithmic predictor and precisely quantify this improvement. We find empirically that although algorithms often outperform their human counterparts *on average*, human judgment can improve algorithmic predictions on *specific* instances (which can be identified ex-ante). In an X-ray classification task, we find that this subset constitutes nearly $30\%$ of the patient population. Our approach provides a natural way of uncovering this heterogeneity and thus enabling effective human-AI collaboration.

## 1 Introduction

Despite remarkable advances in machine learning, human judgment continues to play a critical role in many high-stakes prediction tasks. For example, consider the problem of triage in the emergency room, where healthcare providers assess and prioritize patients for immediate care. On one hand, prognostic algorithms offer significant promise for improving triage decisions; indeed, algorithmic predictions are often more accurate than even expert human decision makers [13, 17, 30, 41, 44, 16, 15, 53]. On the other hand, predictive algorithms may fail to fully capture the relevant context for each individual. For example, an algorithmic risk score may only have access to tabular electronic health records or other structured data (e.g., medical imaging), while a physician has access to many additional modalities—not least of which is the ability to directly examine the patient!

These two observations—that algorithms often outperform humans, but humans often have access to a richer information set—are not in conflict with each other. Indeed, [3] find exactly this phenomenon in an analysis of emergency room triage decisions. This suggests that, even in settings where algorithms outperform humans, algorithms might still benefit from some form of human input. Ideally this collaboration will yield *human-AI complementarity* [6, 61], in which a joint system outperforms either a human or algorithm working alone. Our work thus begins with the following question:

*When (and how) can human judgment improve the predictions of any learning algorithm?*

**Example: X-ray classification.** Consider the problem of diagnosing atelectasis (a partially or fully collapsed lung; we study this task in detail in Section 5). Today's state-of-the-art deep learning models can perform well on these kinds of classification tasks using only a patient's chest X-ray as

38th Conference on Neural Information Processing Systems (NeurIPS 2024).

input [35, 59, 1]. We are interested in whether we can further improve these algorithmic predictions by incorporating a "second opinion" from a physician, particularly because the physician may have access to information (e.g., by directly observing the patient) which is not present in the X-ray.

A first heuristic, without making any assumptions about the available predictive models, is to ask whether a physician can distinguish patients whose imaging data are *identical*. For example, if a physician can correctly indicate that one patient is suffering from atelectasis while another is not—despite the patients having identical chest X-rays—the physician must have information that the X-ray does not capture. In principle, this could form the basis for a statistical test: we could ask whether the physician performs better than random in distinguishing a large number of such patients. If so, even a predictive algorithm which outperforms the physician might benefit from human input.

Of course, we are unlikely to find identical observations in continuous-valued and/or high-dimensional data (like X-rays). A natural relaxation is to instead consider observations which are sufficiently "similar", as suggested by [3]. In this work we propose a more general notion of *algorithmic indistinguishability*, or coarser subsets of inputs in which no algorithm (in some rich, user-defined class) has significant predictive power. We show that these subsets can be discovered via a novel connection to *multicalibration* [33], and formally demonstrate that using human feedback to predict outcomes within these subsets can outperform any algorithmic predictor (in the same user-defined class). In addition to being tractable, this framework is relevant from a decision-theoretic perspective: although we've focused thus far on algorithms' fundamental informational constraints, it is also natural to ask whether an expert provides signal which is merely *difficult* for an algorithm to learn directly (due to e.g., limited training data or computational constraints). Our approach naturally interpolates between these contexts by defining indistinguishability with respect to whichever class of models is practically relevant for a given prediction task. We elaborate on these contributions below.

**Contributions.** We propose a novel framework for human-AI collaboration in prediction tasks. Our approach uses human feedback to refine predictions within sets of inputs which are *algorithmically indistinguishable*, or "look the same" to predictive algorithms. In Section 4 we present a simple method to incorporate this feedback only when it improves on the best feasible predictive model (and precisely quantify this improvement). This extends the "omnipredictors" result of [28] in the special case of squared error, which may be of independent interest.[1] In Section 5 we present experiments demonstrating that although humans fail to outperform algorithmic predictors *on average*, there exist *specific* (algorithmically indistinguishable) instances on which humans are more accurate than the best available predictor (and these instances are identifiable ex ante).[2] In Section 6 we consider the complementary setting in which an algorithm provides recommendations to many downstream users, who independently choose when to comply. We provide conditions under which a predictor is robust to these compliance patterns, and thus be simultaneously optimal for all downstream users.

## 2   Related work

**The relative strengths of humans and algorithms.**   Our work is motivated by large body of literature which studies the relative strengths of human judgment and algorithmic decision making [13, 17, 30, 44] or identifies behavioral biases in decision making [66, 11, 4, 60]. More recent work also studies whether predictive algorithms can *improve* expert decision making [41, 53, 7, 1].

**Recommendations, deferral and complementarity.**   One popular approach for incorporating human judgment into algorithmic predictions is by *deferring* some instances to a human decision maker [49, 58, 52, 37, 55, 38]. Other work studies contexts where human decision makers are free to override algorithmic recommendations [20, 8, 14, 22, 1], which may suggest alternative design criteria for these algorithms [5, 9, 34]. More generally, systems which achieve human-AI *complementarity* (as defined in Section 1) have been previously studied in [2, 5, 69, 23, 64, 19, 47].

[61] develop a comprehensive taxonomy of this area, which generally takes the predictor as given, or learns a predictor which is optimized to complement a particular model of human decision making. In contrast, we give stronger results which demonstrate when human judgment can improve the performance of any model in a rich class of possible predictors (Section 4), or when a single algorithm can complement many heterogeneous users (Section 6).

---

[1]We provide additional technical results in Appendix A. We elaborate on connections to [28] in Appendix D.

[2]Code to replicate our experiments is available at `https://github.com/ralur/heap-repl`.

**Performative prediction.** A recent line of work studies *performative prediction* [57], or settings in which predictions influence future outcomes. For example, predicting the risk of adverse health outcomes may directly inform treatment decisions, which in turn affects future health outcomes. This can complicate the design and evaluation of predictive algorithms, and there is a growing literature which seeks to address these challenges [10, 51, 31, 21, 36, 39, 50, 67, 32, 70, 48, 54, 56]. Performativity is also closely related to the *selective labels problem*, in which some historical outcomes are unobserved as a consequence of past human decisions [45]. Though these issues arise in many canonical human-AI collaboration tasks, we focus on standard supervised learning problems in which predictions do not causally affect the outcome of interest. These include e.g., weather prediction, stock price forecasting and many medical diagnosis tasks, including the X-ray diagnosis task we study in Section 5. In particular, although a physician's diagnosis may inform subsequent treatment decisions, it does not affect the contemporaneous presence or absence of a disease. More generally, our work can be applied to any "prediction policy problem", where accurate predictions can be translated into policy gains without explicitly modeling causality [42].

**Algorithmic monoculture.** Our results can be viewed as one approach to mitigating *algorithmic monoculture*, in which different algorithms make similar decisions and thus similar mistakes [43, 65]. This could occur because these systems are trained on similar datasets, or because they share similar inductive biases. We argue that these are precisely the settings in which a "diversifying" human opinion may be especially valuable. We find empirical evidence for this in Section 5: on instances where multiple models agree on a prediction, human judgment adds substantial predictive value.

**Multicalibration, omnipredictors and boosting.** Our results make use of tools from theoretical computer science, particularly work on *omnipredictors* [28] and its connections to *multicalibration*. [24] show that multicalibration is tightly connected to a cryptographic notion of indistinguishability, which serves as conceptual inspiration for our work. Finally, [27] provide an elegant boosting algorithm for learning multicalibrated partitions that we make use of in our experiments, and [29] provide results which reveal tight connections between a related notion of "swap agnostic learning", multi-group fairness, omniprediction and outcome indistinguishability.

# 3 Methodology and preliminaries

**Notation.** Let $X \in \mathcal{X}$ be a random variable denoting the inputs (or "features") which are available for making algorithmic predictions about an outcome $Y \in [0, 1]$. Let $\hat{Y} \in [0, 1]$ be an expert's prediction of $Y$, and let $x, y, \hat{y}$ denote realizations of the corresponding random variables. Our approach is parameterized by a class of predictors $\mathcal{F}$, which is some set of functions mapping $\mathcal{X}$ to $[0, 1]$. We interpret $\mathcal{F}$ as the class of predictive models which are relevant (or feasible to implement) for a given prediction task; we discuss this choice further below. Broadly, we are interested in whether the expert prediction $\hat{Y}$ provides a predictive signal which cannot be extracted from $X$ by any $f \in \mathcal{F}$.

**Choice of model class $\mathcal{F}$.** For now we place no restrictions on $\mathcal{F}$, but it's helpful to consider a concrete model class (e.g., a specific neural network architecture) from which, given some training data, one could derive a *particular* model (e.g., via empirical risk minimization over $\mathcal{F}$). The choice of $\mathcal{F}$ could be guided by practical considerations; some domains might require interpretable models (e.g., linear functions) or be subject to computational constraints. We might also simply believe that a certain architecture or functional form is well suited to the task of interest. In any case, we are interested in whether human judgment can provide information which is not conveyed by any model in this class, but are agnostic as to *how* this is accomplished: an expert may have information which is not encoded in $X$, or be deploying a decision rule which is not in $\mathcal{F}$ — or both!

Another choice is to take $\mathcal{F}$ to model more abstract limitations on the expert's cognitive process. In particular, to model experts who are subject to "bounded rationality" [63, 40], $\mathcal{F}$ might be the set of functions which can be efficiently computed (e.g., by a circuit of limited complexity). In this case, an expert who provides a prediction which cannot be modeled by any $f \in \mathcal{F}$ must have access to *information* which is not present in the training data. We take the choice of $\mathcal{F}$ as given, but emphasize that these two approaches yield qualitatively different insight about human expertise.

**Indistinguishability with respect to $\mathcal{F}$.** Our approach will be to use human input to distinguish observations which are *indistinguishable* to any predictor $f \in \mathcal{F}$. We formalize this notion of indistinguishability as follows:

**Definition 3.1** ($\alpha$-Indistinguishable subset). For some $\alpha \geq 0$, a set $S \subseteq \mathcal{X}$ is $\alpha$-indistinguishable with respect to a function class $\mathcal{F}$ and target $Y$ if, for all $f \in \mathcal{F}$,

$$|\mathrm{Cov}(f(X), Y \mid X \in S)| \leq \alpha \tag{1}$$

To interpret this definition, observe that the subset $S$ can be viewed as generalizing the intuition given in Section 1 for grouping identical inputs. In particular, rather than requiring that all $x \in S$ are exactly equal, Definition 3.1 requires that all members of $S$ effectively "look the same" for the purposes of making algorithmic predictions about $Y$, as every $f \in \mathcal{F}$ is only weakly related to the outcome within $S$. We now adopt the definition of a multicalibrated partition [28] as follows:

**Definition 3.2** ($\alpha$-Multicalibrated partition). For $K \geq 1$, $S_1 \ldots S_K \subseteq \mathcal{X}$ is an $\alpha$-multicalibrated partition with respect to $\mathcal{F}$ and $Y$ if (1) $S_1 \ldots S_K$ partitions $\mathcal{X}$ and (2) each $S_k$ is $\alpha$-indistinguishable with respect to $\mathcal{F}$ and $Y$.[3]

Intuitively, the partition $\{S_k\}_{k \in [K]}$ "extract[s] all the predictive power" from $\mathcal{F}$ [28]; within each element of the partition, every $f \in \mathcal{F}$ is only weakly related to the outcome $Y$. Thus, while knowing that an input $x$ lies in subset $S_k$ may be highly informative for predicting $Y$ — for example, it may be that $\mathbb{E}[Y \mid X = x] \approx \mathbb{E}[Y \mid X = x']$ for all $x, x' \in S_k$ — no predictor $f \in \mathcal{F}$ provides significant *additional* signal within $S_k$. We provide a stylized example of such partitions in Figure 1 below.

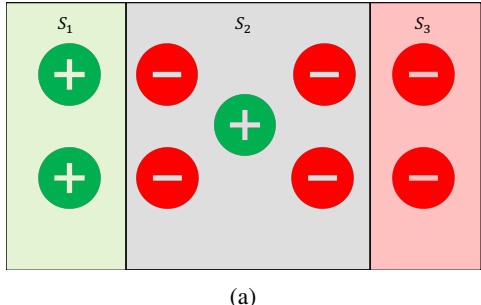 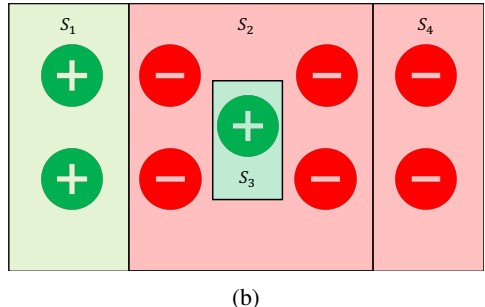

(a) (b)

Figure 1: Partitions which are approximately multicalibrated with respect to the class of hyperplane classifiers (we consider the empirical distribution placing equal probability on each observation). In both panels, no hyperplane classifier has significant discriminatory power within each subset.

It's not obvious that such partitions are feasible to compute, or even that they should exist. We'll show in Appendix B however that a multicalibrated partition can be efficiently computed for many natural classes of functions. Where the relevant partition is clear from context, we use $\mathbb{E}_k[\cdot]$, $\mathrm{Var}_k(\cdot)$, $\mathrm{Cov}_k(\cdot, \cdot)$ to denote expectation, variance and covariance conditional on the event that $\{X \in S_k\}$. For a subset $S \subseteq \mathcal{X}$, we use $\mathbb{E}_S[\cdot]$, $\mathrm{Var}_S(\cdot)$ and $\mathrm{Cov}_S(\cdot, \cdot)$ analogously.

**Incorporating human judgment into predictions.** To incorporate human judgment into predictions, a natural heuristic is to first test whether the conditional covariance $\mathrm{Cov}_k(Y, \hat{Y})$ is nonzero within some indistinguishable subset. Intuitively, this indicates that the expert prediction is informative even though every model $f \in \mathcal{F}$ is not. This suggests a simple method for incorporating human expertise: first, learn a partition which is multicalibrated with respect to $\mathcal{F}$, and then use $\hat{Y}$ to predict $Y$ within each indistinguishable subset. We describe this procedure in Algorithm 1 below, where we define a univariate learning algorithm $\mathcal{A}$ as a procedure which takes one or more $(\hat{y}_i, y_i) \in [0, 1]^2$ training observations and outputs a function which predicts $Y$ using $\hat{Y}$. For example, $\mathcal{A}$ might be an algorithm which fits a univariate linear or logistic regression which predicts $Y$ as a function of $\hat{Y}$.

Algorithm 1 simply learns a different predictor of $Y$ as a function of $\hat{Y}$ within each indistinguishable subset. As we show below, even simple instantiations of this approach can outperform the squared error achieved by *any* $f \in \mathcal{F}$. This approach can also be readily extended to more complicated forms of human input (e.g., freeform text, which can be represented as a high-dimensional vector rather than a point prediction $\hat{Y}$), and can be used to *test* whether human judgment provides information that an algorithm cannot learn from the available training data. We turn to these results below.

---

[3]This is closely related to $\alpha$-approximate multicalibration [28], which requires that Definition 3.1 merely holds in expectation over the partition. We work with a stronger pointwise definition for clarity, but our results can also be interpreted as holding for the 'typical' element of an $\alpha$-approximately multicalibrated partition.

---

**Algorithm 1** A method for incorporating human expertise into algorithmic predictions

---

1: **Inputs:** Training data $\{x_i, y_i, \hat{y}_i\}_{i=1}^n$, a multicalibrated partition $\{S_k\}_{k \in K}$, a univariate regression algorithm $\mathcal{A}$, and a test observation $(x_{n+1}, \hat{y}_{n+1})$
2: **Output:** A prediction of the missing outcome $y_{n+1}$
3: **for** $k = 1$ to $K$ **do**
3:     $Z_k \leftarrow \{(\hat{y}_j, y_j) : x_j \in S_k\}$
3:     $\hat{g}_k \leftarrow \mathcal{A}(Z_k)$
4: **end for**
5: $k^* \leftarrow$ the index of the subset $S_k$ which contains $x_{n+1}$
6: **return** $\hat{g}_{k^*}(\hat{y}_{n+1})$

---

## 4   Technical results

In this section we present our main technical results. For clarity, all results in this section are presented in terms of population quantities, and assume oracle access to a multicalibrated partition. We present corresponding generalization arguments and background on learning multicalibrated partitions in Appendices A and B, respectively. All proofs are deferred to Appendix C.

**Theorem 4.1.** *Let $\{S_k\}_{k \in [K]}$ be an $\alpha$-multicalibrated partition with respect to a model class $\mathcal{F}$ and target $Y$. Let the random variable $J(X) \in [K]$ be such that $J(X) = k$ iff $X \in S_k$. Define $\gamma^*, \beta^* \in \mathbb{R}^K$ as*

$$\gamma^*, \beta^* \in \underset{\gamma \in \mathbb{R}^K, \beta \in \mathbb{R}^K}{\arg\min} \quad \mathbb{E}\left[\left(Y - \gamma_{J(X)} + \beta_{J(X)}\hat{Y}\right)^2\right]. \tag{2}$$

*Then, for any $f \in \mathcal{F}$ and $k \in [K]$,*

$$\mathbb{E}_k\left[\left(Y - \gamma_k^* - \beta_k^*\hat{Y}\right)^2\right] + 4\mathrm{Cov}_k(Y, \hat{Y})^2 \le \mathbb{E}_k\left[\left(Y - f(X)\right)^2\right] + 2\alpha. \tag{3}$$

That is, the squared error incurred by the univariate linear regression of $Y$ on $\hat{Y}$ within each indistinguishable subset outperforms that of any $f \in \mathcal{F}$. This improvement is at least $4\mathrm{Cov}_k(Y, \hat{Y})^2$, up to an additive approximation error $2\alpha$. We emphasize that $\mathcal{F}$ is an arbitrary class, and may include complex, nonlinear predictors. Nonetheless, given a multicalibrated partition, a simple linear predictor can improve on the *best $f \in \mathcal{F}$*. Furthermore, this approach allows us to *selectively* incorporate human feedback: whenever $\mathrm{Cov}_k(Y, \hat{Y}) = 0$, we recover a coefficient $\beta_k^*$ of $0$.[4]

**Nonlinear functions and high-dimensional feedback.** Theorem 4.1 corresponds to instantiating Algorithm 1 with a univariate linear regression, but the same insight generalizes readily to other functional forms. For example, if $Y$ is binary, it might be desirable to instead fit a logistic regression. We provide a similar guarantee for generic nonlinear predictors via Corollary A.1 in Appendix A.

Furthermore, while the results above assume that an expert provides a prediction $\hat{Y} \in [0, 1]$, the same insight extends to richer forms of feedback. For example, in a medical diagnosis task, a physician might produce free-form clinical notes which contain information that is not available in tabular electronic health records. Incorporating this kind of feedback requires a learning algorithm better suited to high-dimensional inputs (e.g., a deep neural network), which motivates our following result.

**Corollary 4.2.** *Let $S$ be an $\alpha$-indistinguishable subset with respect to a model class $\mathcal{F}$ and target $Y$. Let $H \in \mathcal{H}$ denote expert feedback which takes values in some arbitrary domain (e.g., freeform text, which might be tokenized to take values in $\mathbb{Z}^d$ for some $d > 0$), and let $g : \mathcal{H} \to [0, 1]$ be a function which satisfies the following approximate calibration condition for some $\eta \ge 0$ and for all $\beta, \gamma \in \mathbb{R}$:*

$$\mathbb{E}_S[(Y - g(H))^2] \le \mathbb{E}_S[(Y - \gamma - \beta g(H))^2] + \eta. \tag{4}$$

*Then, for any $f \in \mathcal{F}$,*

---

[4]Recall that the population coefficient in a univariate linear regression of $Y$ on $\hat{Y}$ is $\frac{\mathrm{Cov}(Y, \hat{Y})}{\mathrm{Var}(\hat{Y})}$.

$$\mathbb{E}_S\left[(Y - g(H))^2\right] + 4\mathrm{Cov}_S(Y, g(H))^2 \leq \mathbb{E}_S\left[(Y - f(X))^2\right] + 2\alpha + \eta. \tag{5}$$

To interpret this result, notice that (4) requires only that the prediction $g(H)$ cannot be significantly improved by any linear post-processing function. For example, this condition is satisfied by any calibrated predictor $g(H)$.[5] Furthermore, any $g(H)$ which does not satisfy (4) can be transformed by letting $\tilde{g}(H) = \min_{\gamma,\beta} \mathbb{E}[(Y - \gamma - \beta g(H))^2]$; i.e., by linearly regressing $Y$ on $g(H)$, in which case $\tilde{g}(H)$ satisfies (4). This result mirrors Theorem 4.1: a predictor which depends only on human feedback $H$ can improve on the best $f \in \mathcal{F}$ within each element of a multicalibrated partition.

**Testing for informative experts.** While we have thus far focused on incorporating human judgment to improve predictions, we may also be interested in the related question of simply *testing* whether human judgment provides information that cannot be conveyed by any algorithmic predictor. For example, such a test might be valuable in deciding whether to automate a given prediction task.

Theorem 4.1 suggests a heuristic for such a test: if the conditional covariance $\mathrm{Cov}_k(Y, \hat{Y})$ is large, then we might expect that $\hat{Y}$ is somehow "more informative" than any $f \in \mathcal{F}$ within $S_k$. While covariance only measures a certain form of *linear* dependence between random variables, we now show that, in the special case of binary-valued algorithmic predictors, computing the covariance of $Y$ and $\hat{Y}$ within an indistinguishable subset serves as a stronger test for whether $\hat{Y}$ provides *any* predictive information which cannot be expressed by the class $\mathcal{F}$.

**Theorem 4.3.** *Let $\{S_k\}_{k\in[K]}$ be an $\alpha$-multicalibrated partition for a binary-valued model class $\mathcal{F}^{binary}$ and target outcome $Y$. For all $k \in [K]$, let there be $\tilde{f}_k \in \mathcal{F}$ such that $Y \perp\!\!\!\perp \hat{Y} \mid \tilde{f}_k(X), X \in S_k$. Then, for all $k \in [K]$,*

$$\left|\mathrm{Cov}_k(Y, \hat{Y})\right| \leq \sqrt{\frac{\alpha}{2}}. \tag{6}$$

That is, if each indistinguishable subset has a corresponding predictor $\tilde{f}_k$ which "explains" the signal provided by the human, then the covariance of $Y$ and $\hat{Y}$ is bounded within every $S_k$. The contrapositive implies that a sufficiently large value of $\mathrm{Cov}_k(Y, \hat{Y})$ serves as a certificate for the property that *no* $f \in \mathcal{F}$ can fully explain the information that $\hat{Y}$ provides about $Y$ within each indistinguishable subset. This can be viewed as a finer-grained extension of the test proposed in [3].

Taken together, our results demonstrate that algorithmic indistinguishability provides a principled way of reasoning about the complementary value of human judgment. Furthermore, this approach yields a concrete methodology for incorporating this expertise: we can simply use human feedback to predict $Y$ within subsets which are indistinguishable on the basis of $X$ alone. Operationalizing these results depends critically on the ability to *learn* multicalibrated partitions, e.g., via the boosting algorithm proposed in [27]. We provide additional detail on learning such partitions in Appendix B.

## 5 Experiments

### 5.1 Chest X-ray interpretation

We now instantiate our framework in the context of the chest X-ray classification task outlined in Section 1. We consider the eight predictive models studied in [59], which were selected from the leaderboard of a large public competition for X-ray image classification. These models serve as a natural benchmark class $\mathcal{F}$, against which we investigate whether radiologist assessments provide additional predictive value. These models were trained on a dataset of 224,316 chest radiographs collected across 65,240 patients [35], and then evaluated on a holdout set of $500$ randomly sampled radiographs. This holdout set was annotated by eight radiologists for the presence ($Y = 1$) or absence ($Y = 0$) of five selected pathologies; the majority vote of five radiologists serves as a ground truth label, while the remaining three are held out to assess the accuracy of individual radiologists [59].

---

[5]A calibrated predictor is one where $\mathbb{E}_S[Y \mid g(H)] = g(H)$. This is a fairly weak condition; for example, it is satisfied by the constant predictor $g(H) \equiv \mathbb{E}_S[Y]$ [18, 25].

In this section we focus on diagnosing atelectasis (a partial or complete collapse of the lung); we provide results for the other four pathologies in Appendix G. We first show, consistent with [35, 59], that radiologists fail to consistently outperform algorithmic classifiers *on average*. However, we then demonstrate that radiologists do outperform all eight leaderboard algorithms on a large subset (nearly 30% of patients) which is indistinguishable with respect to this class of benchmark predictors. Because radiologists in this experimental setting only have access to the patient's chest X-ray, and because we do not apply any postprocessing to the radiologist assessments (i.e., $\hat{g}_k$, as defined in Algorithm 1, is simply the identity function, which is most natural when $Y$ and $\hat{Y}$ are binary), we interpret these results as providing a lower bound on the improvement that radiologists can provide relative to relying solely on algorithmic classifiers.

**Algorithms are competitive with expert radiologists.** We first compare the performance of the three benchmark radiologists to that of the eight leaderboard algorithms in Figure 2. Following [59], we use the Matthew's Correlation Coefficient (MCC) as a standard measure of binary classification accuracy [12]. The MCC is simply the rescaled covariance between each prediction and the outcome, which corresponds directly to Definition 3.1. In Figure 2 we see that radiologist performance is statistically indistinguishable from that of the algorithmic classifiers.

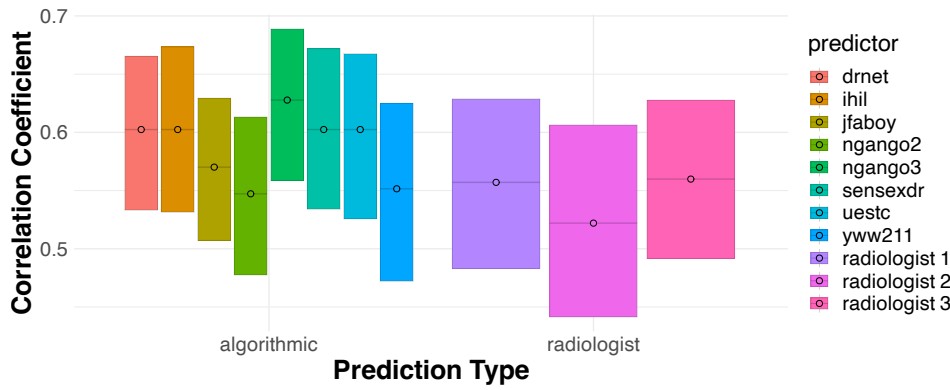

Figure 2: The relative performance of radiologists and predictive algorithms for detecting atelectasis. Each bar plots the Matthews Correlation Coefficient between the corresponding prediction and the ground truth label. Point estimates are reported with 95% bootstrap confidence intervals.

**Radiologists can refine algorithmic predictions.** We now apply the results of Section 4 to investigate *heterogeneity* in the relative performance of humans and algorithms. First, we partition the patients into a pair of approximately indistinguishable subsets, which are exceptionally straightforward to compute when the class $\mathcal{F}$ has a finite number of predictors (we provide additional detail in Appendix F). We plot the conditional performance of both the radiologists and the eight leaderboard algorithms within each of these subsets in Figure 3.

While Figure 2 found no significant differences between radiologists' and algorithms' *overall* performance, Figure 3 reveals a large subset — subset 0, consisting of 29.6% of our sample — where radiologists achieve a better MCC than every algorithm. In particular, every algorithm predicts a positive label for every patient in this subset, and radiologists identify a sizable fraction of true negatives that the algorithms miss. For example, radiologist 1 achieves a true positive rate of 84.0% and a true negative rate of 42.9%, while the algorithms achieve corresponding rates of 100% and 0%.

This partition is not necessarily unique, and in principle an analyst could compare the performance of radiologists and algorithms across different subsets which could yield an even starker difference in conditional performance. However, even for discrete-valued data, searching over all possible subsets is computationally and statistically intractable; instead, our approach provides a principled way of identifying the *particular* subsets in which human judgment is likely to add predictive value.

**Other pathologies.** Although we focus here on atelectasis, and the findings above are consistent for two of the other four pathologies considered in [59] (pleural effusion and consolidation): although radiologists fail to outperform algorithms *on average*, at least two of the radiologists outperform algorithmic predictions on a sizable minority of patients. Results for cardiomegaly and edema appear qualitatively similar, but we lack statistical power. We present these results in Appendix G.

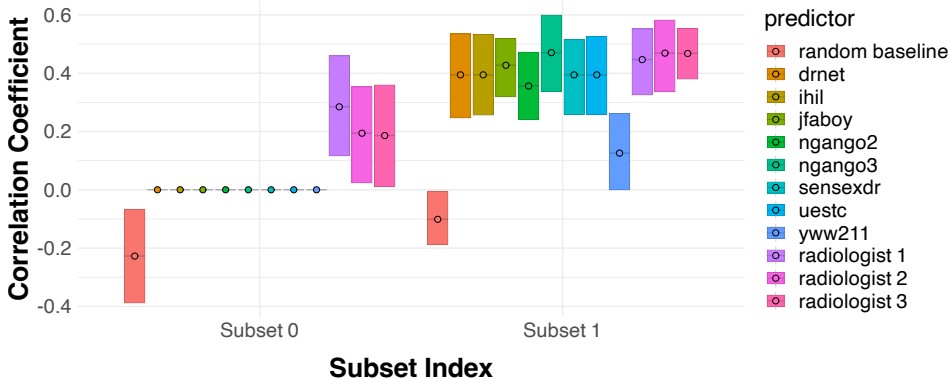

Figure 3: Conditional performance for atelectasis. Within subset $0$ ($n = 148$), all algorithms predict $Y = 1$, thus achieving true positive rate (TPR) $1$, true negative rate (TNR) $0$, and an MCC of $0$. Radiologists achieve a corresponding (TPR, TNR) of $(84.0\%, 42.9\%)$, $(72.6\%, 47.6\%)$ and $(93.4\%, 19.0\%)$, respectively. Subset $1$ ($n = 352$) contains the remaining patients. The baseline is a random permutation of the labels. Confidence intervals for algorithmic performance are not strictly valid (subsets are chosen conditional on the predictions), but are included for reference. All else is as in Figure 2.

## 5.2 Prediction of success in human collaboration

We next consider the visual prediction task studied in [62]. In this work, the authors curate photos taken of participants after they attempt an Escape the Room' puzzle—"a physical adventure game in which a group is tasked with escaping a maze by collectively solving a series of puzzles" [62]. A separate set of subjects are then asked to predict whether the group in each photo succeeded in completing the puzzle. Subjects in the control arm perform this task without any form of training, while subjects in the remaining arms are first provided with four, eight and twelve labeled examples, respectively. Their performance is compared to that of five algorithmic predictors, which use 33 high-level features extracted from each photo (e.g., number of people, gender and ethnic diversity, age distribution etc.) to make a competing prediction. We provide a full list of features in Appendix H.

**Accuracy and indistinguishability in visual prediction.** As in the X-ray diagnosis task, we first compare the performance of human subjects to that of the five off-the-shelf predictive algorithms considered in [62]. We again find that although humans fail to outperform the best predictive algorithms, their predictions add significant predictive value on instances where the algorithms agree on a positive label. As our results are similar to those in the previous section, we defer them to Appendix I. We now use this task to illustrate another feature of our framework, which is the ability to incorporate human judgment into a substantially richer class of models.

**Multicalibration over an infinite class.** While our previous results illustrate that human judgment can complement a small, fixed set of predictive algorithms, it's possible that a richer class could obviate the need for human expertise. To explore this, we now consider an infinitely large but nonetheless simple class of shallow (depth $\leq 5$) regression trees. We denote this class by $\mathcal{F}^{\mathrm{RT5}}$.

As in previous sections, our first step will be to learn a partition which is multicalibrated with respect to $\mathcal{F}^{\mathrm{RT5}}$. However, because $\mathcal{F}^{\mathrm{RT5}}$ is infinitely large, enumerating each $f \in \mathcal{F}^{\mathrm{RT5}}$ and clustering observations according to their predictions is infeasible. Instead, we apply the boosting algorithm proposed in [27] to construct a predictor $h : \mathcal{X} \to [0, 1]$ such that no $f \in \mathcal{F}^{\mathrm{RT5}}$ can substantially improve on the squared error of $h$ within any of its approximate level sets $\{x \mid h(x) = 0\}, \{x \mid h(x) \in (0, .1]\} \ldots \{x \mid h(x) \in [.9, 1]\}$.[6] We plot the correlation of the human subjects' predictions with the true label within these level sets in Figure 4.

Figure 4 highlights a key insight provided by our framework. On one hand, the predictions made by $h$ are more accurate out of sample ($72.2\%$) than even the best performing cohort ($67.3\%$). Nonetheless, the predictions made by all four cohorts of human subjects are predictive of the outcome within every

---

[6]We discuss the connection between this condition and multicalibration in Appendix B. As in the previous section, we cannot necessarily verify whether the multicalibration condition is satisfied empirically. However, the theory provides guidance for choosing subsets, within which we can directly test conditional performance.

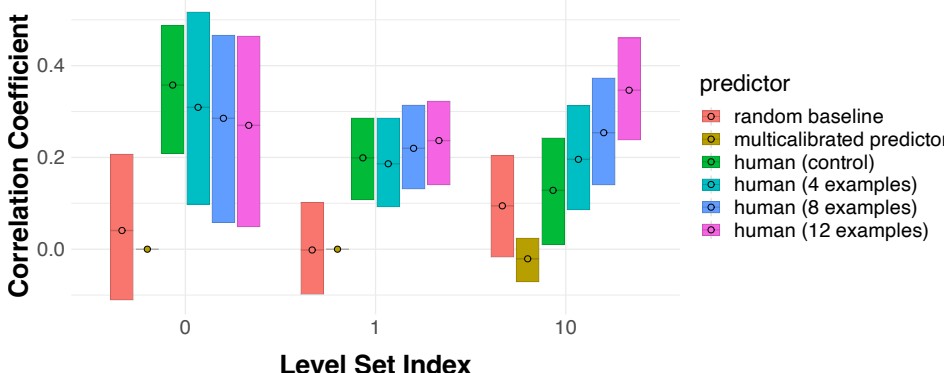

Figure 4: Human performance within the approximate level sets of a predictor $h$ which is multicalibrated over $\mathcal{F}^{\text{RT5}}$. Level sets $0, 1$, and $10$ are the sets $\{x \mid h(x) = 0\}$, $\{x \mid h(x) \in (0, .1]\}$, and $\{x \mid h(x) \in [.9, 1]\}$, and contain $259, 309$ and $292$ observations, respectively. All other level sets are empty in our test set. A random permutation of the labels is included as a baseline.

nonempty level set of of $h$.[7] This suggests that humans provide information which cannot be extracted from the data by *any* $f \in \mathcal{F}^{\text{RT5}}$. While we focus on shallow regression trees for concreteness, this approach extends to any function class for which it is feasible to learn a multicalibrated partition.

# 6 Robustness to noncompliance

We have thus far focused on how an algorithm might incorporate human feedback to improve prediction accuracy, or how an algorithmic decision pipeline might selectively defer to a human expert. However, many decision support tools are deployed in the opposite setting where the *user* instead decides when to defer to *the algorithm*. For example, physicians with access to a prognostic risk score may choose to simply ignore the risk score at their discretion. Furthermore, it is common for hospitals to employ different norms and policies governing the use of algorithmic predictors [46]. Thus, although it is tempting to simply provide all downstream users with the single "best" risk score, such an approach can be suboptimal if users vary in their compliance behavior [5]. We illustrate the challenges of this kind of heterogeneous user behavior via the following stylized example.

**Example: the challenge of noncompliance.** Consider a generic prognostic risk score which makes recommendations regarding patient care. Although physicians generally comply with the algorithm's recommendations, they are free to override it as they see fit. For example, suppose that one particular physician believes (correctly or not) that the algorithm underweights high blood pressure as a risk factor, and thus ignores its recommendations for these patients. A second physician similarly ignores algorithmic recommendations and instead exercises their own judgment for patients 65 and older.

What does an optimal risk score look like in this setting? For the first physician, we would like to select the algorithm which minimizes error on patients who do not have high blood pressure, as these are the patients for whom the physician uses the algorithm's recommendations. Similarly, for the second physician, we would like to minimize error on patients who are under 65. Of course, *there is no guarantee that these are the same algorithm*: empirical risk minimization over the first population will, in general, produce a different predictor than empirical risk minimization over the second population. This is not just a finite sample problem; given any restricted model class (e.g., linear predictors), the optimal predictor for one subpopulation may not be optimal for a different subpopulation. For both practical and ethical reasons however, we cannot design individualized predictors for every physician; we would like to instead provide a risk score which is simultaneously "optimal" (in a sense we make precise below) for every user.

**Noncompliance-robust prediction.** In Appendix A.3, we show that, without further assumptions, the setting described above poses a statistically intractable problem: if physicians choose whether

---

[7]Though this result may initially seem counterintuitive, recall the classical Simpson's paradox: in a setting where $\hat{Y}$ is uncorrelated (or even negatively correlated) with $Y$, there may still exist a partition of the data such that the two are positively correlated within *every* subgroup.

to comply in arbitrary ways, then we need a predictor which is simultaneously optimal for every possible patient subpopulation. The only predictor which satisfies this criterion is the Bayes optimal predictor, which is infeasible to learn in a finite data regime.

However, suppose instead that physicians decide whether to defer to the algorithm using relatively simple heuristics. If we believe we can model these heuristics as a "simple" function of observable patient characteristics — e.g., that all compliance patterns can be expressed as a shallow decision tree, even if particular compliance behavior varies across physicians — then we can leverage this structure to design a single optimal predictor. In particular, we show next that, given a partition which is multicalibrated over the class of possible user compliance patterns, we can learn predictors which remain optimal even when users only selectively adopt the algorithm's recommendations.

**Theorem 6.1.** *Let $\Pi$ be a class of binary compliance policies, where, for $\pi \in \Pi$, $\pi(x) = 1$ indicates that the user complies with the algorithm at $X = x$. Let $\mathcal{F}$ be a class of predictors and let $\{S_k\}_{k \in [K]}$ be a partition which is $\alpha$-multicalibrated with respect to $\Pi$ and the product class $\{f(X)\pi(X) \mid f \in \mathcal{F}, \pi \in \Pi\}$. Then, $\forall f \in \mathcal{F}, \pi \in \Pi, k \in [K]$,*

$$\mathbb{E}_k[(Y - \mathbb{E}_k[Y])^2 \mid \pi(X) = 1] \le \mathbb{E}_k[(Y - f(X))^2 \mid \pi(X) = 1] + \frac{6\alpha}{\mathbb{P}_k(\pi(X) = 1)}. \tag{7}$$

That is, given an appropriately multicalibrated partition, we can derive a predictor which is simultaneously near-optimal for every downstream user. In particular, observe that the left hand side of (7) is the squared error incurred by the constant prediction $\mathbb{E}_k[Y]$ within $S_k$ *when the user defers to the algorithm*. Although this prediction does not depend on the policy $\pi$, it remains competitive with the squared error incurred by any $f \in \mathcal{F}$ for *any policy*. Unsurprisingly, the bound becomes vacuous as $\mathbb{P}_k(\pi(X) = 1)$ goes to 0 (we cannot hope to learn anything on arbitrarily rare subsets). This is consistent with our interpretation of $\pi$ however, as the performance of the algorithm matters little if the decision maker ignores nearly all recommendations.

This result is complementary to those in Section 4—rather than learning to incorporate feedback from a single expert, we can instead learn a single predictor which is (nearly) optimal for a rich class of downstream users whose behavior is modeled by some $\pi \in \Pi$.

## 7    Discussion and limitations

In this work we propose an indistinguishability-based framework for human-AI collaboration. Under this framework, we develop a set of methods for testing whether experts provide a predictive signal which cannot be replicated by an algorithmic predictor, and extend our results to settings in which users selectively adopt algorithmic recommendations. Beyond these methodological contributions, we argue that our framing clarifies *when* and *why* human judgment can improve algorithmic performance. In particular, a primary theme in our work is that even if humans do not consistently outperform algorithms on average, *selectively* incorporating human judgment can often improve predictions.

A key limitation of our work is a somewhat narrow focus on minimizing a well-defined loss function over a well-defined (and stationary) distribution. This fails to capture decision makers with richer, multidimensional preferences (e.g., fairness, robustness or simplicity), and does not extend to settings in which predictions *influence* future outcomes (see the discussion of performative prediction in Section 2) or the distribution otherwise changes over time. However, we view indistinguishability as a powerful primitive for modeling these more complex scenarios; for example, a decision maker might impose additional preferences — like a desire for some notion of fairness — to distinguish inputs which are otherwise indistinguishable with respect to the "primary" outcome of interest. At a technical level, our results rely on the ability to efficiently *learn* multicalibrated partitions. While we give conditions under which this is feasible in Appendix B and a finite sample analysis in Appendix A, finding such partitions can be challenging for rich function classes.

Finally, we caution that even in contexts which fit neatly into our framework, human decision makers can be critical for ensuring interpretability and accountability. Thus, although our approach can provide guidance for choosing the appropriate level of automation, it does not address the practical or ethical concerns which arise. Despite these limitations, we argue that indistinguishability helps to clarify the role of human expertise in algorithmic decision making, and this framing in turn provides fundamental conceptual and methodological insights for enabling effective human-AI collaboration.

## Acknowledgements

RA, MR and DS are supported in part by a Stephen A. Schwarzman College of Computing Seed Grant. DS is supported in part by an NSF FODSI Award (2022448).

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

# A  Additional technical results

In this section we present additional technical results which complement those in the main text. All proofs are deferred to Appendix C.

## A.1   A nonlinear analog of Theorem 4.1

Below we provide a simple extension of Theorem 4.1 from univariate linear regression to arbitrary univariate predictors of $Y$ given $\hat{Y}$.

**Corollary A.1.** *Let $S$ be an $\alpha$-indistinguishable subset with respect to a model class $\mathcal{F}$ and target $Y$. Let $g : [0,1] \to [0,1]$ be a function which satisfies the following approximate Bayes-optimality condition for $\eta \geq 0$:*

$$\mathbb{E}_S[(Y - g(\hat{Y}))^2] \leq \mathbb{E}_S[(Y - \mathbb{E}_S[Y \mid \hat{Y}])^2] + \eta. \tag{8}$$

*Then, for any $f \in \mathcal{F}$,*

$$\mathbb{E}_S\left[(Y - g(\hat{Y}))^2\right] + 4\mathrm{Cov}_S(Y,\hat{Y})^2 \leq \mathbb{E}_S\left[(Y - f(X))^2\right] + 2\alpha + \eta. \tag{9}$$

That is, any function $g$ which is nearly as accurate (in terms of squared error) as the univariate conditional expectation function $\mathbb{E}_S[Y \mid \hat{Y}]$ provides the same guarantee as in Theorem 4.1. This conditional expectation function is exactly what e.g., a logistic regression of $Y$ on $\hat{Y}$ seeks to model. We provide a proof in Appendix C.

## A.2   A finite sample analog of Corollary 4.2

For simplicity, the technical results in Section 4 are presented in terms of population quantities. In this section we consider the empirical analogue of Corollary 4.2, and provide a generalization argument which relates these empirical quantities to the corresponding population results in Section 4. We focus our attention on Corollary 4.2, as the proof follows similarly for Theorem 4.1 and Corollary A.1.

Let $\mathcal{G}$ be some class of predictors mapping $\mathcal{H}$ to $[0,1]$. We'll begin with the setup of Corollary 4.2, with $S \subseteq \mathcal{X}$ denoting a fixed, measurable subset of the input space (we'll generalize to a full partition $S_1 \ldots S_K$ below). Further let $n_S \equiv \sum_{i=1}^{n} \mathbb{1}(x_i \in S)$ denote the number of training examples which lie in the subset $S \subseteq \mathcal{X}$, and $\{y_i, h_i\}_{i=1}^{n_S}$ denote i.i.d. samples from the unknown joint distribution over the random variables $(Y, H)$ conditional on the event that $X \in S$. Let $\hat{g}_S \equiv \arg\min_{g \in \mathcal{G}} \frac{1}{n_S} \sum_{i=1}^{n_S} (y_i - g(h_i))^2$ denote the empirical risk minimizer within $S$. Our goal will be to show that if there exists some $g_S^* \in \mathcal{G}$ satisfying (4), then the empirical risk minimizer $\hat{g}_S$ also approximately satisfies (4) with high probability.

**Lemma A.2.** *Let $\mathcal{L} = \{\ell_g : g \in \mathcal{G}\}$, where $\ell_g(x,y) \equiv (y - g(x))^2$, denote the class of squared loss functions indexed by $g \in \mathcal{G}$, and let $R_{n_S}(\mathcal{L})$ denote the Rademacher complexity of $\mathcal{L}$. Let $P(S) \equiv \mathbb{P}(X \in S)$ denote the measure of $S$. Then, for any $\delta \geq 0$, with probability at least $(1 - e^{-\frac{nP(S)\delta^2}{4}})(1 - 2e^{-P(S)n})$, we have*

$$\mathbb{E}_S[\ell_{\hat{g}}] \leq \mathbb{E}_S[\ell_{g^*}] + 4R_{n_S}(\mathcal{L}) + 2\delta, \quad and \quad n_S \geq nP(S)/2. \tag{10}$$

That is, if there exists some $g^* \in \mathcal{G}$ satisfying (4), then the empirical risk minimizer $\hat{g} \in \mathcal{G}$ within the subset $S$ also satisfies (4) with high probability, up to an approximation error that depends on the Rademacher complexity of $\mathcal{L}$. For many natural classes of functions, including linear functions, the Rademacher complexity (1) tends to 0 as $n \to \infty$ and (2) can be sharply bounded in finite samples (see e.g., Chapter 4 in [68]). We provide a proof of Lemma A.2 in Appendix C.

**Extending Lemma A.2 to a partition of $\mathcal{X}$.** The result above is stated for a single subset $S$, and depends critically on the *measure* of that subset $P(S) \equiv \mathbb{P}(X \in S)$. We now show this generalization argument can be extended to a partition of the input space $S_1 \ldots S_K \subseteq \mathcal{X}$ by arguing that Lemma A.2 applies to "most" instances sampled from the marginal distribution over $\mathcal{X}$. Specifically, we show that the probability that $X$ lies in *any* subset $S$ with measure $P(S)$ approaching $0$ is a low probability event; for the remaining inputs, Lemma A.2 applies.

**Corollary A.3.** *Let $\{S_k\}_{k \in [K]}$ be a (not necessarily multicalibrated) partition of the input space, chosen independently of the training data, and let $n_k$ denote the number of training observations which lie in a given subset $S_k \subseteq \mathcal{X}$. Then, for any $\epsilon > 0$ and $\delta \geq 0$, $X$ lies in a subset $S_k \subseteq \mathcal{X}$ such that,*

$$\mathbb{E}_k[\ell_{\hat{g}}] \leq \mathbb{E}_k[\ell_{g^*}] + 4R_{n_k}(\mathcal{L}) + 2\delta \tag{11}$$

*with probability at least $(1 - \epsilon)(1 - e^{\frac{-n_k \epsilon \delta^2}{4K}})(1 - 2e^{-\frac{n_k \epsilon}{K}})$ over the distribution of the training data $\{x_i, y_i, \hat{y}_i\}_{i=1}^n$ and a test observation $(x_{n+1}, y_{n+1}, \hat{y}_{n+1})$.*

The preceding corollary indicates that Lemma A.2 also holds for a "typical" subset of the input space, replacing the dependence on the measure of any given subset $P(S)$ with a lower bound on the probability that a test observation $x_{n+1}$ lies in some subset whose measure is at least $\frac{\epsilon}{K}$. We provide a formal proof in Appendix C.

## A.3 The impossibility of arbitrary deferral policies

In this section we formalize the argument in Section 6 to show that it is infeasible to learn predictors which are simultaneously "optimal" (in a sense we make precise below) for many downstream users who independently choose when to comply with the algorithm's recommendations. We provide a proof in Appendix C.

**Lemma A.4.** *Let $\mathcal{F}$ be some class of predictors which map a countable input space $\mathcal{X}$ to $[0, 1]$. We interpret a compliance policy $\pi : \mathcal{X} \to [0, 1]$ such that $\pi(x) = 1$ indicates that the user complies with the algorithm's recommendation at $X = x$. For all $f \in \mathcal{F}$, unless $f = \mathbb{E}[Y \mid X]$ almost everywhere, then there exists a deferral policy $\pi : \mathcal{X} \to \{0, 1\}$ and constant $c \in [0, 1]$ such that:*

$$\mathbb{E}[(Y - f(X))^2 \mid \pi(X) = 1] > \mathbb{E}[(Y - c)^2 \mid \pi(X) = 1] \tag{12}$$

Lemma A.4 indicates that for any predictor $f$ which is not the Bayes optimal predictor, there exists a compliance policy which causes it to underperform a constant prediction on the instances for which it is ultimately responsible. Because learning the Bayes optimal predictor from a finite sample of data is generally infeasible, this indicates that a predictor cannot reasonably be made robust to an arbitrary deferral policy. The proof, which we provide below, is intuitive: the decision maker can simply choose to comply on exactly those instances where $f$ performs poorly.

# B Learning multicalibrated partitions

In this section we discuss two sets of conditions on $\mathcal{F}$ which enable the efficient computation of multicalibrated partitions. An immediate implication of our first result is that any class of Lipschitz predictors induce a multicalibrated partition.

**Level sets of $\mathcal{F}$ are multicalibrated.** Observe that one way in which Definition 3.1 is trivially satisfied (with $\alpha = 0$) is whenever every $f \in \mathcal{F}$ is *constant* within a subset $S \subseteq \mathcal{X}$. We relax this insight as follows: if the variance of every $f \in \mathcal{F}$ is bounded within $S$, then $S$ is approximately indistinguishable with respect to $\mathcal{F}$.

**Lemma B.1.** *Let $\mathcal{F}$ be a class of predictors and $S \subseteq \mathcal{X}$ be a subset of the input space. If:*

$$\max_{f \in \mathcal{F}} \text{Var}(f(X) \mid X \in S) \leq 4\alpha^2, \tag{13}$$

*then $S$ is $\alpha$-indistinguishable with respect to $\mathcal{F}$ and $Y$.*

This result yields a natural corollary: the approximate level sets of $\mathcal{F}$ (i.e., sets in which the range of every $f \in \mathcal{F}$ is bounded) are approximately indistinguishable. We state this result formally as Corollary B.2 below. We use exactly this approach to finding multicalibrated partitions in our study of a chest X-ray classification task in Section 5.

**Corollary B.2.** *Let $\mathcal{F}$ be a class of predictors whose range is bounded within some $S \subseteq \mathcal{X}$. That is, for all $f \in \mathcal{F}$:*

$$\max_{x \in S} f(x) - \min_{x' \in S} f(x') \leq 4\alpha \tag{14}$$

*Then $S$ is $\alpha$-indistinguishable with respect to $\mathcal{F}$.*

Lemma B.1 also implies a simple algorithm for finding multicalibrated partitions when $\mathcal{F}$ is Lipschitz with respect to some distance metric $d : \mathcal{X} \times \mathcal{X} \to \mathbb{R}$: observations which are close under $d(\cdot, \cdot)$ are guaranteed to be approximately indistinguishable with respect to $\mathcal{F}$. We state this result formally as Corollary B.3 below.

**Corollary B.3.** *Let $\mathcal{F}^{Lip(L,d)}$ be the set of $L$-Lipschitz functions with respect to some distance metric $d(\cdot, \cdot)$ on $\mathcal{X}$. That is:*

$$|f(x) - f(x')| \leq Ld(x, x') \ \forall \ f \in \mathcal{F}^{Lip(L,d)} \tag{15}$$

*Let $\{S_k\}_{k \in K}$ for $K \subseteq \mathbb{N}$ be some $(4\alpha/L)$-net on $\mathcal{X}$ with respect to $d(\cdot, \cdot)$. Then $\{S_k\}_{k \in K}$ is $\alpha$-multicalibrated with respect to $\mathcal{F}^{Lip(L,d)}$.*

Proofs of the results above are provided in Appendix C.

**Multicalibration via boosting.** Recent work by [27] demonstrates that multicalibration is closely related to *boosting* over a function class $\mathcal{F}$. In this section we first provide conditions, adapted from [27], which imply that the level sets of a certain predictor $h : \mathcal{X} \to [0, 1]$ are multicalibrated with respect to $\mathcal{F}$; that is, the set $\{x \mid h(x) = v\}$ for every $v$ in the range of $h$ is approximately indistinguishable. We then discuss how these conditions yield a natural boosting algorithm for *learning* a predictor $h$ which induces a multicalibrated partition. In the lemma below, we use $\mathcal{R}(f)$ to denote the range of a function $f$.

**Lemma B.4.** *Let $\mathcal{F}$ be a function class which is closed under affine transformations; i.e., $f \in \mathcal{F} \Rightarrow a + bf \in \mathcal{F}$ for all $a, b \in \mathbb{R}$, and let $\tilde{\mathcal{F}} = \{f \in \mathcal{F} \mid \mathcal{R}(f) \subseteq [0, 1]\}$. Let $Y \in [0, 1]$ be the target outcome, and $h : \mathcal{X} \to [0, 1]$ be some predictor with countable range $\mathcal{R}(h) \subseteq [0, 1]$. If, for all $f \in \mathcal{F}, v \in \mathcal{R}(h)$:*

$$\mathbb{E}\left[(h(X) - Y)^2 - (f(X) - Y)^2 \mid h(X) = v\right] < \alpha^2, \tag{16}$$

*then the level sets of $h$ are $(2\alpha)$-multicalibrated with respect to $\tilde{\mathcal{F}}$ and $Y$.*

To interpret this result, observe that (16) is the difference between the mean squared error of $f$ and the mean squared error of $h$ within each level set $S_v = \{x \in \mathcal{X} \mid h(x) = v\}$. Thus, if the best $f \in \mathcal{F}$ fails to significantly improve on the squared error of $h$ within a given level set $S_v$, then $S_v$ is indistinguishable with respect to $\tilde{\mathcal{F}}$ (which is merely $\mathcal{F}$ restricted to functions that lie in $[0, 1]$). [27] give a boosting algorithm which, given a squared error regression oracle[8] for $\mathcal{F}$, outputs a predictor $h$ which satisfies (16). We make use of this algorithm in Section 5 to learn a partition of the input space in a visual prediction task. Although the class we consider there (the class of shallow regression trees $\mathcal{F}^{RT5}$) is not closed under affine transformations, boosting over this class captures the spirit of our main result: while no $f \in \mathcal{F}^{RT5}$ can improve accuracy within the level sets of $h$, humans provide additional predictive signal within three of them.

Taken together, the results in this section demonstrate that multicalibrated partitions can be efficiently computed for many natural classes of functions, which in turn enables the application of results in Section 4.

---

[8]Informally, a squared error regression oracle for $\mathcal{F}$ is an algorithm which can efficiently output $\arg\min_{f \in \mathcal{F}} \mathbb{E}[(Y - f(X))^2]$ for any distribution over $X, Y$. When the distribution is over a finite set of training data, this is equivalent to empirical risk minimization. We refer to [27] for additional details, including generalization arguments.

## C Proofs of primary results

In this section we present proofs of our main results. Proofs of auxiliary lemmas are deferred to Appendix E.

### C.1 Omitted proofs from Section 4

**Lemma C.1.** *The following simple lemma will be useful in our subsequent proofs. Let $X \in \{0, 1\}$ be a binary random variable. Then for any other random variable, $Y$:*

$$\text{Cov}(X, Y) \tag{17}$$
$$= \mathbb{P}(X = 1)\left(\mathbb{E}[Y \mid X = 1] - \mathbb{E}[Y]\right) \tag{18}$$
$$= \mathbb{P}(X = 0)\left(\mathbb{E}[Y] - \mathbb{E}[Y \mid X = 0]\right) \tag{19}$$

*This is exactly corollary 5.1 in [28]. We provide the proof in Appendix E.*

**Proof of Theorem 4.1**

*Proof.* A well known fact about univariate linear regression is that the coefficient of determination (or $r^2$) is equal to the square of the Pearson correlation coefficient between the regressor and the outcome (or $r$). In our context, this means that within any indistinguishable subset $S_k$ we have:

$$1 - \frac{\mathbb{E}_k\left[\left(Y - \gamma_k^* - \beta_k^* \hat{Y}\right)^2\right]}{\mathbb{E}_k\left[\left(Y - \mathbb{E}_k[Y]\right)^2\right]} = \frac{\text{Cov}_k(Y, \hat{Y})^2}{\text{Var}_k(Y)\text{Var}_k(\hat{Y})} \tag{20}$$

$$\Rightarrow \mathbb{E}_k\left[\left(Y - \mathbb{E}_k[Y]\right)^2\right] - \mathbb{E}_k\left[\left(Y - \gamma_j^* - \beta_j^* \hat{Y}\right)^2\right] = \frac{\text{Cov}_k(Y, \hat{Y})^2}{\text{Var}(\hat{Y})} \tag{21}$$

$$\Rightarrow \mathbb{E}_k\left[\left(Y - \gamma_j^* - \beta_j^* \hat{Y}\right)^2\right] = \mathbb{E}_k\left[\left(Y - \mathbb{E}_k[Y]\right)^2\right] - \frac{\text{Cov}_k(Y, \hat{Y})^2}{\text{Var}(\hat{Y})} \tag{22}$$

$$\leq \mathbb{E}_k\left[\left(Y - \mathbb{E}_k[Y]\right)^2\right] - 4\text{Cov}_k(Y, \hat{Y})^2 \tag{23}$$

Where (23) is an application of Popoviciu's inequality for variances, and makes use of the fact that $\hat{Y} \in [0, 1]$ almost surely. We can then obtain the final result by applying the following lemma, which extends the main result in [28]. We provide a proof in Appendix E, but for now simply state the result as Lemma C.2 below.

**Lemma C.2.** *Let $\{S_k\}_{k \in [K]}$ be an $\alpha$-multicalibrated partition with respect to a real-valued function class $\mathcal{F} = \{f : \mathcal{X} \to [0, 1]\}$ and target outcome $Y \in [0, 1]$. For all $f \in \mathcal{F}$ and $k \in [K]$, it follows that:*

$$\mathbb{E}_k\left[\left(Y - \mathbb{E}[Y]\right)^2\right] \leq \mathbb{E}_k\left[\left(Y - f(X)\right)^2\right] + 2\alpha \tag{24}$$

We provide further discussion of the relationship between Lemma C.2 and the main result of [28] in Appendix D below.

Chaining inequalities (24) and (23) yields the final result:

$$\mathbb{E}_k\left[\left(Y - \gamma_j^* - \beta_j^* \hat{Y}\right)^2\right] \leq \mathbb{E}_k\left[\left(Y - f(X)\right)^2\right] + 2\alpha - 4\text{Cov}_k(Y, \hat{Y})^2 \ \forall \ f \in \mathcal{F} \tag{25}$$

$$\Rightarrow \mathbb{E}_k\left[\left(Y - \gamma_j^* - \beta_j^* \hat{Y}\right)^2\right] + 4\text{Cov}_k(Y, \hat{Y})^2 \leq \mathbb{E}_k\left[\left(Y - f(X)\right)^2\right] + 2\alpha \ \forall \ f \in \mathcal{F} \tag{26}$$

$$\square$$

**Proof of Corollary 4.2**

*Proof.* The proof is almost immediate. Let $\gamma^*, \beta^* \in \mathbb{R}$ be the population regression coefficients obtained by regressing $Y$ on $g(H)$ within $S$ (as in Theorem 4.1; the only difference is that we consider a single indistinguishable subset rather than a multicalibrated partition). This further implies, by the approximate calibration condition (4):

$$\mathbb{E}_S\left[(Y - g(H))^2\right] \leq \mathbb{E}_S\left[(Y - \gamma_k^* - \beta_k^* g(H))^2\right] + \eta \tag{27}$$

The proof then follows from that of Theorem 4.1, replacing $\hat{Y}$ with $g(H)$.

$\square$

**Proof of Theorem 4.3**

*Proof.* Fix any $k \in [K]$.

$$\left|\mathrm{Cov}_k(Y, \hat{Y})\right| \tag{28}$$

$$= \left|\mathbb{E}_k[\mathrm{Cov}_k(Y, \hat{Y} \mid \tilde{f}_k(X)] + \mathrm{Cov}_k(\mathbb{E}_k[Y \mid \tilde{f}_k(X)], \mathbb{E}_k[\hat{Y} \mid \tilde{f}_k(X)])\right| \tag{29}$$

$$= \left|\mathrm{Cov}_k(\mathbb{E}[Y \mid \tilde{f}_k(X)], \mathbb{E}_k[\hat{Y} \mid \tilde{f}_k(X)])\right| \tag{30}$$

$$\leq \sqrt{\mathrm{Var}(\mathbb{E}_k[Y \mid \tilde{f}_k(X)])\mathrm{Var}_k(\mathbb{E}[\hat{Y} \mid \tilde{f}_k(X)])} \tag{31}$$

$$\leq \frac{1}{2}\sqrt{\mathrm{Var}_k(\mathbb{E}_k[Y \mid \tilde{f}_k(X)])} \tag{32}$$

Where (29) is the law of total covariance, (30) follows from the assumption that $Y \perp\!\!\!\perp \hat{Y} \mid \tilde{f}_k(X), X \in S_k$, (31) is the Cauchy-Schwarz inequality and (32) applies Popoviciu's inequality to bound the variance of $\mathbb{E}[\hat{Y} \mid \tilde{f}_k(X)]$ (which is assumed to lie in $[0, 1]$ almost surely).

We now focus on bounding $\mathrm{Var}_k(\mathbb{E}_k[Y \mid \tilde{f}_k(X)])$. Recall that by assumption, $|\mathrm{Cov}_k(Y, \tilde{f}_k(X))| \leq \alpha$, so we should expect that conditioning on $\tilde{f}_k(X)$ does not change the expectation of $Y$ by too much.

$$\mathrm{Var}_k(\mathbb{E}_k[Y \mid \tilde{f}_k(X)]) \tag{33}$$

$$= \mathbb{E}_k[(\mathbb{E}_k[Y \mid \tilde{f}_k(X)] - \mathbb{E}_k[\mathbb{E}_k[Y \mid \tilde{f}_k(X)]])^2] \tag{34}$$

$$= \mathbb{E}_k[(\mathbb{E}_k[Y \mid \tilde{f}_k(X)] - \mathbb{E}_k[Y])^2] \tag{35}$$

$$= \mathbb{P}_k(\tilde{f}_k(X) = 1)(\mathbb{E}_k[Y \mid \tilde{f}_k(X) = 1] - \mathbb{E}_k[Y])^2$$
$$+ \mathbb{P}_k(\tilde{f}_k(X) = 0)(\mathbb{E}_k[Y \mid \tilde{f}_k(X) = 0] - \mathbb{E}_k[Y])^2 \tag{36}$$

$$\leq \mathbb{P}_k(\tilde{f}_k(X) = 1)\left|\mathbb{E}_k[Y \mid \tilde{f}_k(X) = 1] - \mathbb{E}_k[Y]\right|$$
$$+ \mathbb{P}_k(\tilde{f}_k(X) = 0)\left|\mathbb{E}_k[Y \mid \tilde{f}_k(X) = 0] - \mathbb{E}_k[Y]\right| \tag{37}$$

Where the last step follows because $Y$ is assumed to be bounded in $[0, 1]$ almost surely. Applying Lemma C.1 to (37) yields:

$$\mathrm{Var}_k(\mathbb{E}_k[Y \mid \tilde{f}_k(X)]) \leq \left|2\mathrm{Cov}_k(Y, \tilde{f}_k(X))\right| \leq 2\alpha \tag{38}$$

Where the second inequality follows because our analysis is conditional on $X \in S_k$ for some $\alpha$-indistinguishable subset $S_k$. Plugging (38) into (32) completes the proof.

$\square$

## C.2 Omitted proofs from Section 6

**Proof of Lemma A.4**

*Proof.* Let $f \in \mathcal{F}$ be any model and let $S \subseteq \mathcal{X}$ be a subset such that (1) $\mathbb{P}_X(S) > 0$ (2) the Bayes optimal predictor $\mathbb{E}[Y \mid X]$ is constant within $S$ and (3) $f(X) \neq \mathbb{E}[Y \mid X = x]$ for all $x \in S$. Such a subset must exist by assumption. It follows immediately that choosing

$$\pi(x) = \begin{cases} 1 \text{ if } x \in S \\ 0 \text{ otherwise} \end{cases}$$

suffices to ensure that $f(X)$ underperforms the constant prediction $c_S = \mathbb{E}[Y \mid X \in S]$ on the subset which $\pi$ delegates to $f$. This implies that even if $\mathcal{F}$ includes the class of constant predictors $\{f(X) = c \mid c \in \mathbb{R}\}$—perhaps the simplest possible class of predictors—then we cannot hope to find some $f^* \in \mathcal{F}$ which is simultaneously optimal for any choice of deferral policy. $\square$

**Proof of Theorem 6.1**

*Proof.* We start with the assumption that $\{S_k\}_{k \in [K]}$ is $\alpha$-multicalibrated with respect to $\Pi$ and the product class $\{f(X)\pi(X) \mid f \in \mathcal{F}, \pi \in \Pi\}$. That is, both of the following hold:

$$|\text{Cov}_k(Y, \pi(X))| \leq \alpha \; \forall \; \pi \in \Pi, k \in [K] \tag{39}$$
$$|\text{Cov}_k(Y, f(X)\pi(X))| \leq \alpha \; \forall \; f \in \mathcal{F}, \pi \in \Pi, k \in [K] \tag{40}$$

First, we'll show that this implies that the covariance of $Y$ and $f(X)$ is bounded even conditional on compliance. To streamline presentation we state this as a separate lemma; the proof is provided further below.

**Lemma C.3.** *Given the setup of Theorem 6.1, the following holds for all $k \in [K], f \in \mathcal{F}$ and $\pi \in \Pi$:*

$$|\text{Cov}_k(Y, f(X) \mid \pi(X) = 1)| \leq \frac{2\alpha}{\mathbb{P}_k(\pi(X) = 1)} \tag{41}$$

*We provide a proof in Appendix E. By Lemma C.2, Lemma C.3 implies, for all $k \in [K], f \in \mathcal{F}$ and $\pi \in \Pi$:*

$$\mathbb{E}_k \left[ (Y - \mathbb{E}_k[Y \mid \pi(X) = 1])^2 \mid \pi(X) = 1 \right]$$
$$\leq \mathbb{E}_k \left[ (Y - f(X))^2 \mid \pi(X) = 1 \right] + \frac{4\alpha}{\mathbb{P}(\pi(X) = 1)} \tag{42}$$

*This is close to what we want to prove, except that the prediction $E_k[Y \mid \pi(X) = 1]$ depends on the choice of the policy $\pi(\cdot)$. We'll argue that by (39), $\mathbb{E}_k[Y \mid \pi(X) = 1] \approx \mathbb{E}_k[Y]$. Indeed, because $\pi(\cdot)$ is binary, we can apply Lemma C.1 to recover:*

$$|\text{Cov}(\pi(X), Y)| = \mathbb{P}_k(\pi(X) = 1) |\mathbb{E}_k[Y \mid \pi(X) = 1] - \mathbb{E}_k[Y]| \tag{43}$$
$$\Rightarrow |\mathbb{E}_k[Y \mid \pi(X) = 1] - \mathbb{E}_k[Y]| \leq \frac{\alpha}{\mathbb{P}_k(\pi(X) = 1)} \tag{44}$$

*We rewrite the LHS of (42) to make use of this identity as follows:*

$$\mathbb{E}_k \left[ (Y - \mathbb{E}_k[Y \mid \pi(X) = 1])^2 \mid \pi(X) = 1 \right] \tag{45}$$
$$= \mathbb{E}_k \left[ ((Y - \mathbb{E}_k[Y]) + (\mathbb{E}_k[Y] - \mathbb{E}_k[Y \mid \pi(X) = 1]))^2 \mid \pi(X) = 1 \right] \tag{46}$$

$$= \mathbb{E}_k\Big[\Big((Y - \mathbb{E}_k[Y])^2 + (\mathbb{E}_k[Y] - \mathbb{E}_k[Y \mid \pi(X) = 1])^2$$
$$+ 2(Y - \mathbb{E}_k[Y])(\mathbb{E}_k[Y] - \mathbb{E}_k[Y \mid \pi(X) = 1])\Big) \mid \pi(X) = 1\Big] \tag{47}$$

$$\geq \mathbb{E}_k\left[\left((Y - \mathbb{E}_k[Y])^2 + 2(Y - \mathbb{E}_k[Y])(\mathbb{E}_k[Y] - \mathbb{E}_k[Y \mid \pi(X) = 1])\right) \mid \pi(X) = 1\right] \tag{48}$$

$$= \mathbb{E}_k\left[(Y - \mathbb{E}_k[Y])^2 \mid \pi(X) = 1\right]$$
$$+ 2\left(\mathbb{E}_k[Y] - \mathbb{E}_k[Y \mid \pi(X) = 1]\right)\left(\mathbb{E}_k[Y \mid \pi(X) = 1] - \mathbb{E}_k[Y]\right) \tag{49}$$

$$\geq \mathbb{E}_k\left[(Y - \mathbb{E}_k[Y])^2 \mid \pi(X) = 1\right] - \frac{2\alpha}{\mathbb{P}_k(\pi(X) = 1)} \tag{50}$$

*Where the last step follows by observing that either (1) $\mathbb{E}_k[Y] = \mathbb{E}_k[Y \mid \pi(X) = 1]$ or (2) exactly one of $(\mathbb{E}_k[Y] - \mathbb{E}_k[Y \mid \pi(X) = 1])$ or $(\mathbb{E}_k[Y \mid \pi(X) = 1] - \mathbb{E}_k[Y])$ is strictly positive. Assume that $\mathbb{E}_k[Y] \neq \mathbb{E}_k[Y \mid \pi(X) = 1]$; otherwise the bound follows trivially. We bound the positive term by recalling that $Y$ lies in $[0, 1]$, and we bound the negative term by applying (44). Thus, the product of these two terms is at least $\frac{-\alpha}{\mathbb{P}_k(\pi(X)=1)}$. Finally, combining (50) with (42) completes the proof.*

$\square$

### C.3 Omitted proofs from Appendix A

**Proof of Corollary A.1**

*Proof.* Observe that, because the conditional expectation function $\mathbb{E}_k[Y \mid \hat{Y}]$ minimizes squared error with respect to all univariate functions of $\hat{Y}$, we must have:

$$\mathbb{E}_S\left[(Y - \mathbb{E}_S[Y \mid \hat{Y}])^2\right] \leq \mathbb{E}_S\left[(Y - \gamma^* - \beta^*\hat{Y})^2\right] \tag{51}$$

Where $\gamma^* \in \mathbb{R}$, $\beta^* \in \mathbb{R}$ are the population regression coefficients obtained by regression $Y$ on $g(H)$ as in Theorem 4.1. This further implies, by the approximate Bayes-optimality condition (8):

$$\mathbb{E}_S\left[(Y - g(\hat{Y}))^2\right] \leq \mathbb{E}_S\left[(Y - \gamma_k^* - \beta_k^*\hat{Y})^2\right] + \eta \tag{52}$$

The proof then follows immediately from that of Theorem 4.1.

$\square$

**Proof of Lemma A.2**

*Proof.* We will adopt the notation from the setup of Corollary 4.2. Further let $\mathcal{G}$ be some class of predictors mapping $\mathcal{H}$ to $[0, 1]$, over which we seek to learn the mean-squared-error minimizing $g_S^* \in \mathcal{G}$ within some subset $S \subseteq \mathcal{X}$.

Let $n_S \equiv \sum_{i=1}^n \mathbb{1}(x_i \in S)$ denote the number of training examples which lie in the subset $S$, and let $\widehat{\mathbb{E}}_S[\ell_g] \equiv \frac{1}{n_S}\sum_{i=1}^{n_S}(Y_i - g(X_i))^2$ denote the empirical loss incurred by some $g \in \mathcal{G}$ within $S$. Finally, let $\mathbb{E}_S[\ell_g] \equiv E[(Y - g(X))^2 \mid X \in S]$ denote the population analogue of $\widehat{\mathbb{E}}_S[\ell_g]$.

By Hoeffding's inequality we have:

$$\mathbb{P}\left(n_S \geq \frac{nP(S)}{2}\right) \geq \left(1 - 2e^{-P(S)n}\right) \tag{53}$$

that is, $n_S$ is at least half its expectation with high probability. Let $\mathcal{L} = \{\ell_g : g \in \mathcal{G}\}$, where $\ell_g(x, y) = (y - g(x))^2$, be the class of squared loss functions indexed by $g \in \mathcal{G}$. Let $R_{n_S}(\mathcal{L})$ denote the Rademacher complexity of $\mathcal{L}$, which is defined as follows:

**Definition C.4.** Rademacher Complexity

For a fixed $n \in \mathbb{N}$, let $\epsilon_1 \ldots \epsilon_n$ denote $n$ i.i.d. Rademacher random variables (recall that Rademacher random variables take values 1 and $-1$ with equal probability $\frac{1}{2}$). Let $Z_1 \ldots Z_n$ denote i.i.d. random variables taking values in some abstract domain $\mathcal{Z}$, and let $\mathcal{T}$ be a class of real-valued functions over the domain $\mathcal{Z}$. The Rademacher complexity of $\mathcal{T}$ is denoted by $R_n(\mathcal{T})$, and is defined as follows:

$$R_n(\mathcal{T}) = \mathbb{E}\left[\sup_{t \in \mathcal{T}} \left| \frac{1}{n} \sum_{i=1}^{n} \epsilon_i t(Z_i) \right|\right] \tag{54}$$

Where the expectation is taken over both $\epsilon_1 \ldots \epsilon_n$ and $Z_1 \ldots Z_n$. Intuitively, the Rademacher complexity is the expected maximum correlation between some $t \in \mathcal{T}$ and the noise vector $\epsilon_1 \ldots \epsilon_n$.

We now make use of a standard uniform convergence result, which is stated in terms of the Rademacher complexity of a function class. We reproduce this theorem (lightly adapted to our notation) from the textbook treatment provided in [68] below:

**Theorem C.5.** *(adapted from [68]) For any $b$-uniformly-bounded class of functions $\mathcal{T}$, any positive integer $n \geq 1$ and any scalar $\delta \geq 0$, we have:*

$$\sup_{t \in \mathcal{T}} \left| \frac{1}{n} \sum_{i=1}^{n} t(Z_i) - \mathbb{E}[t(Z_1)] \right| \leq 2\mathcal{R}_n(\mathcal{T}) + \delta \tag{55}$$

*with probability at least $1 - \exp\left(-\frac{n\delta^2}{2b^2}\right)$.*

Applying Theorem C.5 (noting that $\mathcal{L}$ is uniformly bounded in $[0, 1]$) implies:

$$\sup_{g \in \mathcal{G}} \left| \widehat{\mathbb{E}}_S[\ell_g] - \mathbb{E}_S[\ell_g] \right| \leq 2R_{n_S}(\mathcal{L}) + \delta \tag{56}$$

with probability at least $1 - e^{\frac{-n_S\delta^2}{2}}$. Finally, combining (53) with (56) further implies, for any $\delta \geq 0$,

$$\mathbb{E}_S[\ell_{\hat{g}}] \leq \mathbb{E}_S[\ell_{g^*}] + 4R_{n_S}(\mathcal{L}) + 2\delta \tag{57}$$

with probability at least $(1 - e^{-\frac{nP(S)\delta^2}{4}})(1 - 2e^{-P(S)n})$, as desired.

**Proof of Corollary A.3**

*Proof.* We'll adopt the notation from the setup of Lemma A.2 and Corollary A.3. Observe that, if we were to take the union of every element of the partition $\{S_k\}_{k \in K}$ with measure $P(S) \equiv \mathbb{P}(X \in S_k) \leq \frac{\epsilon}{K}$, the result would be a subset of the input space with measure at most $K \times \frac{\epsilon}{K} = \epsilon$. Note that this is merely an analytical device; we need not identify which subsets these are.

Thus, with probability at least $(1 - \epsilon)$, a newly sampled test observation will lie in some element of the partition $\{S_k\}_{k \in [K]}$ with measure at least $\frac{\epsilon}{K}$. Conditional on this event, we can directly apply the result of Lemma A.2, plugging in a lower bound of $\frac{\epsilon}{K}$ for $P(S)$. This yields, for any $\epsilon > 0$ and $\delta \geq 0$, $X$ lies in a subset $S_k \subseteq \mathcal{X}$ such that,

$$\mathbb{E}_k[\ell_{\hat{g}}] \leq \mathbb{E}_k[\ell_{g^*}] + 4R_{n_k}(\mathcal{L}) + 2\delta \tag{58}$$

with probability at least $(1 - \epsilon)(1 - e^{\frac{-n\epsilon\delta^2}{4K}})(1 - 2e^{-\frac{n\epsilon}{K}})$ over the distribution of the training data $\{x_i, y_i, \hat{y}_i\}_{i=1}^{n}$ and a test observation $(x_{n+1}, y_{n+1}, \hat{y}_{n+1})$, as desired. $\qquad \square$

$\square$

### C.4 Omitted proofs from Appendix B

**Proof of Lemma B.1**

*Proof.* We want to show $|\mathrm{Cov}(Y, f(X) \mid X \in S)| \leq \alpha$ for all $f \in \mathcal{F}$ and some $S$ such that $\max_{f \in \mathcal{F}} \mathrm{Var}(f(X) \mid X \in S) \leq 4\alpha^2$.

Fix any $f \in \mathcal{F}$. We then have:

$$|\mathrm{Cov}(Y, f(X) \mid X \in S)| \tag{59}$$

$$\leq \sqrt{\mathrm{Var}(Y \mid X \in S)\mathrm{Var}(f(X) \mid X \in S)} \tag{60}$$

$$\leq \sqrt{\frac{1}{4} \times \mathrm{Var}(f(X) \mid X \in S)} \tag{61}$$

$$\leq \sqrt{\frac{1}{4} \times 4\alpha^2} \tag{62}$$

$$= \alpha \tag{63}$$

Where (60) is the Cauchy-Schwarz inequality, (61) is Popoviciu's inequality and makes use of the fact that $Y$ is bounded in $[0, 1]$ by assumption, and (62) uses the assumption that $\max_{f \in \mathcal{F}} \mathrm{Var}(f(X) \mid X \in S) \leq 4\alpha^2$.

$\square$

**Proof of Corollary B.2**

*Proof.* We want to show that $\forall~f \in \mathcal{F}$:

$$|\mathrm{Cov}(Y, f(X) \mid X \in S)| \leq \alpha \tag{64}$$

By assumption, $f(X)$ is bounded in a range of $4\alpha$ within $S$. From this it follows by Popoviciu's inequality for variances that $\forall~f \in \mathcal{F}$:

$$\mathrm{Var}(f(X) \mid X \in S_j) \leq \frac{(4\alpha)^2}{4} = 4\alpha^2 \tag{65}$$

The proof then follows from Lemma B.1.

$\square$

**Proof of Corollary B.3**

*Proof.* We want to show that $\forall~f \in \mathcal{F}^{\mathrm{Lip}(L,d)}, k \in K$:

$$|\mathrm{Cov}_k(Y, f(X))| \leq \alpha \tag{66}$$

Because $S_k$ is part of a $4\alpha/L$-net, there exists some $m \in [0, 1]$ such that $\mathbb{P}(f(X) \in [m, m + 4\alpha] \mid X \in S_k) = 1$; that is, $f(X)$ is bounded almost surely in some interval of length $4\alpha$. From this it follows by Popoviciu's inequality for variances that $\forall~f \in \mathcal{F}^{\mathrm{Lip}(L,d)}, k \in K$:

$$\mathrm{Var}_k(f(X)) \leq \frac{(4\alpha)^2}{4} = 4\alpha^2 \tag{67}$$

The remainder of the proof follows from Lemma B.1.

$\square$

**Proof of Lemma B.4**

*Proof.* The result follows Lemma 3.3 and Lemma 6.8 in [27]. We provide a simplified proof below, adapted to our notation. We'll use $\mathbb{E}_v[\cdot]$ to denote the expectation conditional on the event that $\{h(X) = v\}$ for each $v \in \mathcal{R}(h)$. We use $\mathrm{Cov}_v(\cdot, \cdot)$ analogously.

Our proof will proceed in two steps. First we'll show that:

$$\forall v \in \mathcal{R}(h), f \in \mathcal{F}, \mathbb{E}_v[(h(X) - Y)^2 - (f(X) - Y)^2] < \alpha^2 \tag{68}$$

$$\Rightarrow \mathbb{E}_v[f(X)(Y - v)] < \alpha \ \forall \ v \in \mathcal{R}(h), f \in \tilde{\mathcal{F}} \tag{69}$$

This condition states that if there does not exist some $v$ in the range of $h$ where the best $f \in \mathcal{F}$ improves on the squared error incurred by $h$ by more than $\alpha^2$, then the predictor $h(\cdot)$ is $\alpha$-multicalibrated in the sense of [27] with respect to the constrained class $\tilde{\mathcal{F}}$. We then show that the level sets of a predictor $h(\cdot)$ which satisfies (69) form a multicalibrated partition (Definition 3.2). That is:

$$\mathbb{E}_v[f(X)(Y - v)] \leq \alpha \ \forall v \in \mathcal{R}(h), f \in \tilde{\mathcal{F}} \ \Rightarrow \mathrm{Cov}_v(f(X), Y) \leq 2\alpha \ \forall v \in \mathcal{R}(h), f \in \tilde{\mathcal{F}} \tag{70}$$

That is, the level sets $S_v = \{x \mid h(x) = v\}$ form a $(2\alpha)$-multicalibrated partition with respect to $\tilde{\mathcal{F}}$.

First, we'll prove the contrapositive of (69). This proof is adapted from that of Lemma 3.3 in [27]. Suppose there exists some $v \in \mathcal{R}(h)$ and $f \in \tilde{\mathcal{F}}$ such that

$$\mathbb{E}_v[f(X)(Y - v)] \geq \alpha \tag{71}$$

Then there exists $f' \in \mathcal{F}$ such that:

$$\mathbb{E}_v[(f'(X) - Y)^2 - (h(X) - Y)^2] \geq \alpha^2 \tag{72}$$

Proof: let $\eta = \frac{\alpha}{\mathbb{E}_v[f(X)^2]}$ and $f' = v + \frac{\alpha}{\mathbb{E}_v[f(X)^2]} f(X) = v + \eta f(X)$. Then:

$$\mathbb{E}_v\left[(h(X) - Y)^2 - (f'(X) - Y)^2\right] \tag{73}$$

$$= \mathbb{E}_v\left[(v - Y)^2 - (v + \eta f(X) - Y)^2\right] \tag{74}$$

$$= \mathbb{E}_v\left[v^2 + Y^2 - 2Yv - v^2 - \eta^2 f(X)^2 - Y^2 - 2v\eta f(X) + 2vY + 2\eta f(X)Y\right] \tag{75}$$

$$= \mathbb{E}_v\left[2\eta f(X)(Y - v) - \eta^2 f(X)^2\right] \tag{76}$$

$$= \mathbb{E}_v\left[2\eta f(X)(Y - v)\right] - \frac{\alpha^2}{\mathbb{E}_v[f(X)^2]} \tag{77}$$

$$\geq 2\eta\alpha - \frac{\alpha^2}{\mathbb{E}_v[f(X)^2]} \tag{78}$$

$$= \frac{\alpha^2}{\mathbb{E}_v[f(X)^2]} \tag{79}$$

$$\geq \alpha^2 \tag{80}$$

Where the last step follows because we took $f \in \tilde{\mathcal{F}}$, the subset of the function class $\mathcal{F}$ which only takes values in $[0, 1]$. This implies that if instead $\mathbb{E}_v[(f'(X) - Y)^2 - (h(X) - Y)^2] < \alpha^2$ for all $v \in \mathcal{R}(h), f' \in \mathcal{F}$, then $\mathbb{E}_v[f(X)(Y - v)] < \alpha$ for all $v \in \mathcal{R}(h)$ and $f \in \tilde{\mathcal{F}}$. Next we prove (70); that is, $\mathbb{E}_v[f(X)(Y - v)] < \alpha$ for all $v \in \mathcal{R}(h)$ and $f \in \tilde{\mathcal{F}}$ implies $|\mathrm{Cov}_v(f(X), Y)| \leq 2\alpha$ for all $v \in \mathcal{R}(h), f \in \tilde{\mathcal{F}}$.

The proof is adapted from that of Lemma 6.8 in [27]; our proof differs beginning at (88). Fix some $f \in \tilde{\mathcal{F}}$ and $v \in \mathcal{R}(h)$. By assumption we have, for all $v \in \mathcal{R}(h)$ and $f \in \tilde{\mathcal{F}}$,

$$\mathbb{E}_v[f(X)(Y - v)] < \alpha \tag{81}$$

Then we can show:

$$|\text{Cov}_v(f(X), Y)| \tag{82}$$
$$= |\mathbb{E}_v[f(X)Y] - \mathbb{E}_v[f(X)]\mathbb{E}_v[Y]| \tag{83}$$
$$= |\mathbb{E}_v[f(X)Y] - \mathbb{E}_v[f(X)]\mathbb{E}_v[Y] + v\mathbb{E}_v[f(X)] - v\mathbb{E}_v[f(X)]| \tag{84}$$
$$= |\mathbb{E}_v[f(X)(Y - v)] + \mathbb{E}_v[f(X)](v - \mathbb{E}_v[Y])| \tag{85}$$
$$\leq |\mathbb{E}_v[f(X)(Y - v)]| + |\mathbb{E}_v[f(X)](v - \mathbb{E}_v[Y])| \tag{86}$$
$$= |\mathbb{E}_v[f(X)(Y - v)]| + |\mathbb{E}_v[f(X)](\mathbb{E}_v[Y] - v)| \tag{87}$$
$$\leq \alpha + |\mathbb{E}_v[f(X)](\mathbb{E}_v[Y] - v)| \tag{88}$$

Where the last step follows from the assumption (81). Now, let $f'(X) \equiv \mathbb{E}_v[f(X)]$ be the constant function which takes the value $\mathbb{E}_v[f(X)]$. We can write (88) as follows:

$$\alpha + |\mathbb{E}_v[f(X)](\mathbb{E}_v[Y] - v)| = \alpha + |f'(X)(\mathbb{E}_v[Y] - v)| \tag{89}$$
$$= \alpha + |\mathbb{E}_v[f'(X)(Y - v)]| \tag{90}$$

Because $\mathcal{F}$ is closed under affine transformations, it contains all constant functions, and thus, $f'(X) \in \mathcal{F}$. $\tilde{\mathcal{F}}$, by definition, is the subset of $\mathcal{F}$ whose range lies in $[0, 1]$. Because $f \in \tilde{\mathcal{F}}$, it must be that $\mathbb{E}_v[f(X)] \in [0, 1]$ and thus, $f' \in \tilde{\mathcal{F}}$. So, we can again invoke (81) to show:

$$\alpha + |\mathbb{E}_v[f'(X)(Y - v)]| \leq 2\alpha \tag{91}$$

Which completes the proof.

$\square$

## D    Relating Lemma C.2 to Omnipredictors [28]

In this section we compare Lemma C.2 to the main result of [28]. While the main result of [28] applies broadly to convex, Lipschitz loss functions, we focus on the special case of minimizing squared error. In this case, we show that Lemma C.2 extends the main result of [28] to cover real-valued outcomes under somewhat weaker and more natural conditions. We proceed in three steps: first, to provide a self-contained exposition, we state the result of [28] for real-valued outcomes in the special case of squared error (Lemma D.1 and Lemma D.2 below). Second, we derive a matching bound using Lemma C.2 (our result), which we do by demonstrating that the conditions of Lemma D.2 imply the conditions of Lemma C.2. Finally, we show that Lemma C.2 applies in more generality than Lemma D.2, under conditions which match those of Definition 3.2.

We first state the main result of [28] (adapted to our notation) below, which holds for binary outcomes $Y \in \{0, 1\}$.[9]

**Lemma D.1** (Omnipredictors for binary outcomes, specialized to squared error ([28], Theorem 6.3))**.** *Let $S$ be a subset which is $\alpha$-indistinguishable with respect to a real-valued function class $\mathcal{F}$ and a binary target outcome $Y \in \{0, 1\}$. Then, for all $f \in \mathcal{F}$,*

$$\mathbb{E}_S\left[(Y - \mathbb{E}[Y])^2\right] \leq \mathbb{E}_S\left[(Y - f(X))^2\right] + 4\alpha \tag{92}$$

---

[9]As discussed in Section 1, we also continue to elide the distinction between the 'approximate' multicalibration of [28] and our focus on individual indistinguishable subsets. The results in this section can again be interpreted as holding for the 'typical' element of an approximately multicalibrated partition.

This result makes use of the fact that for any fixed $y \in [0, 1]$, the squared error function is 2-Lipschitz with respect to $f(x)$ over the interval $[0, 1]$. This is similar to Lemma C.2, but requires that $Y$ is binary-valued. In contrast, Lemma C.2 allows for real-valued $Y \in [0, 1]$, and gains a factor of 2 on the RHS.[10] [28] provide an alternate extension of Lemma D.1 to bounded, real-valued $Y$, which we present below for comparison to Lemma C.2.

**Extending Lemma D.1 to real-valued** $Y$. Fix some $\epsilon > 0$, and let $B(\epsilon) = \{0, 1, 2 \ldots \lfloor \frac{2}{\epsilon} \rfloor\}$. Let $\tilde{Y}$ be a random variable which represents a discretization of $Y$ into bins of size $\frac{\epsilon}{2}$. That is, $\tilde{Y} = \min_{b \in B(\epsilon)} |Y - \frac{b\epsilon}{2}|$. Let $\mathcal{R}(\tilde{Y})$ denote the range of $\tilde{Y}$. Observe that the following holds for any function $g : \mathcal{X} \to [0, 1]$:

$$\left| \mathbb{E}[(\tilde{Y} - g(X))^2] - \mathbb{E}[(Y - g(X))^2] \right| \leq \epsilon \tag{93}$$

Where (93) follows because the function $(y - g(x))^2$ is 2-Lipschitz with respect to $g(x)$ over $[0, 1]$ for all $y \in [0, 1]$. We now work with the discretization of $\tilde{Y}$, and provide an analogue to Lemma D.1 under a modified indistinguishability condition for discrete-valued $\tilde{Y}$, which we'll show is stronger than Definition 3.1.

**Lemma D.2** (Extending Lemma D.1 to real-valued $Y$ ([28], adapted from Theorem 8.1)). *Let $\mathcal{R}(f)$ denote the range of a function $f$, and let $1(\cdot)$ denote the indicator function. Let $S$ be a subset of the input space $\mathcal{X}$ which satisfies the following condition with respect to a function class $\mathcal{F}$ and discretized target $\tilde{Y}$:*

*For all $f \in \mathcal{F}$ and $\tilde{y} \in \mathcal{R}(\tilde{Y})$, if:*

$$\left| \mathrm{Cov}_S(1(\tilde{Y} = \tilde{y}), f(X)) \right| \leq \alpha \tag{94}$$

*Then:*

$$\mathbb{E}_S \left[ \left( \tilde{Y} - \mathbb{E}_S[\tilde{Y}] \right)^2 \right] \leq \mathbb{E}_S \left[ \left( \tilde{Y} - f(X) \right)^2 \right] + 2 \left\lceil \frac{2}{\epsilon} \right\rceil \alpha \tag{95}$$

To interpret this result, observe that (95) yields a bound which is similar to Lemma D.1 under a modified 'pointwise' indistinguishability condition (94) for any discretization $\tilde{Y}$ of $Y$. Combining (95) with (93) further implies:

$$\mathbb{E}_S \left[ \left( Y - \mathbb{E}_S[\tilde{Y}] \right)^2 \right] \leq \mathbb{E}_S \left[ (Y - f(X))^2 \right] + 2 \left\lceil \frac{2}{\epsilon} \right\rceil \alpha + 2\epsilon \tag{96}$$

**Deriving Lemma D.2 using Lemma C.2**

We show next that the 'pointwise' condition (94) for $\alpha \geq 0$ implies our standard indistinguishability condition (Definition 3.1) for $\alpha' = \lceil \frac{2}{\epsilon} \rceil \alpha$. This will allow us to apply Lemma C.2 to obtain a bound which is identical to (96). Thus, we show that Lemma C.2 is at least as general as Lemma D.2.

**Lemma D.3.** *Let $S$ be a subset satisfying (94). Then, for all $f \in \mathcal{F}$,*

$$\left| \mathrm{Cov}_S(\tilde{Y}, f(X)) \right| \leq \left\lceil \frac{2}{\epsilon} \right\rceil \alpha \tag{97}$$

We provide a proof in Appendix E. Thus, combining assumption (94) with Lemma C.2 and (93) recovers a result which is identical to Lemma D.2. That is, for all $f \in \mathcal{F}$:

---

[10]Note that Lemma C.2 also requires that each $f \in \mathcal{F}$ takes values in $[0, 1]$, but this is without loss of generality when the outcome is bounded in $[0, 1]$; projecting each $f \in \mathcal{F}$ onto $[0, 1]$ can only reduce squared error.

$$\left| \mathrm{Cov}_S(1(\tilde{Y} = \tilde{y}), f(X)) \right| \leq \alpha \tag{98}$$

$$\Rightarrow \left| \mathrm{Cov}_S(\tilde{Y}, f(X)) \right| \leq \left\lceil \frac{2}{\epsilon} \right\rceil \alpha \tag{99}$$

$$\Rightarrow \mathbb{E}_S \left[ \left( \tilde{Y} - \mathbb{E}_S[\tilde{Y}] \right)^2 \right] \leq \mathbb{E}_S \left[ \left( \tilde{Y} - f(X) \right)^2 \right] + 2 \left\lceil \frac{2}{\epsilon} \right\rceil \alpha \tag{100}$$

$$\Rightarrow \mathbb{E}_S \left[ \left( Y - \mathbb{E}[\tilde{Y}] \right)^2 \right] \leq \mathbb{E}_S \left[ (Y - f(X))^2 \right] + 2 \left\lceil \frac{2}{\epsilon} \right\rceil \alpha + 2\epsilon \tag{101}$$

Where (99) follows from Lemma D.3, (100) follows from Lemma C.2 and (101) follows from (93).

**Extending Lemma C.2 beyond Lemma D.2**

Finally, to show that Lemma C.2 extends Lemma D.2, it suffices to provide a distribution over $f(X)$ for some $f \in \mathcal{F}$ and a discrete-valued $\tilde{Y}$ taking $l \geq 1$ values such that Definition 3.1 is satisfied at level $\alpha \geq 0$, but (94) is not satisfied at $\alpha' = (\alpha/l)$ (though in fact that taking $\alpha' = \alpha$ also suffices for the following counterexample).

Consider the joint distribution in which the events $\{\tilde{Y} = 0, f(X) = 0\}$, $\{\tilde{Y} = \frac{1}{2}, f(X) = \frac{1}{2}\}$ and $\{\tilde{Y} = \frac{1}{2}, f(X) = 1\}$ occur with equal probability $\frac{1}{3}$ conditional on $\{X \in S\}$ for some $S \subseteq \mathcal{X}$. We suppress the conditioning event $\{X \in S\}$ for clarity. Then:

$$\mathrm{Cov}(1(\tilde{Y} = 0), f(X)) = \mathbb{P}(\tilde{Y} = 1) \left( \mathbb{E}[f(X) \mid \tilde{Y} = 0] - \mathbb{E}[f(X)] \right) = -\frac{1}{6} \tag{102}$$

On the other hand we have:

$$\mathrm{Cov}(\tilde{Y}, f(X)) = \mathbb{E}[\tilde{Y} f(X)] - \mathbb{E}[\tilde{Y}] \mathbb{E}[f(X)] \tag{103}$$

$$= \mathbb{E}[\tilde{Y} \mathbb{E}[f(X) \mid \tilde{Y}]] - \mathbb{E}[\tilde{Y}] \mathbb{E}[f(X)] \tag{104}$$

$$= \left( \frac{1}{3} \times 0 + \frac{2}{3} \times \frac{1}{2} \times \frac{3}{4} \right) - \frac{1}{3} \times \frac{1}{2} = \frac{1}{12} \tag{105}$$

That is, we have $\left| \mathrm{Cov}(\tilde{Y}, f(X)) \right| = \frac{1}{12} < 3 \left| \mathrm{Cov}(1(\tilde{Y} = 0), f(X)) \right| = \frac{1}{2}$. Thus, Lemma C.2 establishes a result which is similar to (D.2) for real-valued $Y$ under the weaker and more natural condition that $|\mathrm{Cov}(Y, f(X))|$ is bounded, which remains well-defined for real-valued $Y$, rather than requiring the stronger pointwise bound (94) for some discretization $\tilde{Y}$.

Finally, we briefly compare Lemma C.2 to Theorem 8.3 in [28], which generalizes Lemma D.2 to hold for linear combinations of the functions $f \in \mathcal{F}$ and to further quantify the gap between the 'canonical predictor' $\mathbb{E}_k[Y]$ and any $f \in \mathcal{F}$ (or linear combinations thereof). These extensions are beyond the scope of our work, but we briefly remark that the apparently sharper bound of Theorem 8.3 is due to an incorrect assumption that the squared loss $(y - g(x))^2$ is 1-Lipschitz with respect to $g(x)$ over the interval $[0, 1]$, for any $y \in [0, 1]$. Correcting this to a Lipschitz constant of 2 recovers the same bound as (101).

# E   Proofs of auxiliary lemmas

**Proof of Lemma C.1**

*Proof.* We'll first prove (18).

$$\mathrm{Cov}(X, Y) = \mathbb{E}[XY] - \mathbb{E}[X]\mathbb{E}[Y] \tag{106}$$

$$= \mathbb{E}[\mathbb{E}[XY \mid X]] - \mathbb{E}[X]\mathbb{E}[Y] \tag{107}$$

$$= \mathbb{P}(X = 1)\mathbb{E}[XY \mid X = 1] + \mathbb{P}(X = 0)\mathbb{E}[XY \mid X = 0] - \mathbb{E}[X]\mathbb{E}[Y] \tag{108}$$
$$= \mathbb{P}(X = 1)\mathbb{E}[Y \mid X = 1] - \mathbb{E}[X]\mathbb{E}[Y] \tag{109}$$
$$= \mathbb{P}(X = 1)\mathbb{E}[Y \mid X = 1] - \mathbb{P}(X = 1)\mathbb{E}[Y] \tag{110}$$
$$= \mathbb{P}(X = 1)\left(\mathbb{E}[Y \mid X = 1] - \mathbb{E}[Y]\right) \tag{111}$$

As desired. To prove (19), let $X' = 1 - X$. Applying the prior result yields:

$$\mathrm{Cov}(X', Y) = \mathbb{P}(X' = 1)\left(\mathbb{E}[Y \mid X' = 1] - \mathbb{E}[Y]\right) \tag{112}$$

Because $X' = 1 \Leftrightarrow X = 0$, it follows that:

$$\mathrm{Cov}(X', Y) = \mathbb{P}(X = 0)\left(\mathbb{E}[Y \mid X = 0] - \mathbb{E}[Y]\right) \tag{113}$$

Finally, because covariance is a bilinear function, $\mathrm{Cov}(X', Y) = \mathrm{Cov}(1 - X, Y) = -\mathrm{Cov}(X, Y)$. Chaining this identity with (113) yields the result. $\qquad\square$

**Proof of Lemma C.2**

The result we want to prove specializes Theorem 6.3 in [28] to the case of squared error, but our result allows $Y \in [0, 1]$ rather than $Y \in \{0, 1\}$. The first few steps of our proof thus follow that of Theorem 6.3 in [28]; our proof diverges starting at (118). We provide a detailed comparison of these two results in Appendix D above.

*Proof.* Fix any $k \in [K]$. We want to prove the following bound:

$$\mathbb{E}_k[(Y - \mathbb{E}_k[Y])^2] \leq \mathbb{E}_k[(Y - f(X))^2] + 4\alpha \tag{114}$$

It suffices to show instead that:

$$\mathbb{E}_k[(Y - \mathbb{E}_k[f(X)])^2] \leq \mathbb{E}_k[(Y - f(X))^2] + 4\alpha \tag{115}$$

From this the result follows, as $\mathbb{E}_k[(Y - \mathbb{E}_k[Y])^2] \leq \mathbb{E}_k[(Y - c)^2]$ for any constant $c$. To simplify notation, we drop the subscript $k$ and instead let the conditioning event $\{X \in S_k\}$ be implicit throughout. We first show:

$$\mathbb{E}[(Y - f(X))^2] = \mathbb{E}\left[\mathbb{E}\left[(Y - f(X))^2 \mid Y\right]\right] \geq \mathbb{E}\left[(Y - \mathbb{E}[f(X) \mid Y])^2\right] \tag{116}$$

Where the second inequality is an application of Jensen's inequality (the squared loss is convex in $f(X)$). From this it follows that:

$$\mathbb{E}\left[(Y - \mathbb{E}[f(X)])^2\right] - \mathbb{E}\left[(Y - f(X))^2\right] \tag{117}$$
$$\leq \mathbb{E}\left[(Y - \mathbb{E}[f(X)])^2 - (Y - \mathbb{E}[f(X) \mid Y])^2\right] \tag{118}$$
$$= \mathbb{E}\left[\mathbb{E}[f(X)]^2 - 2Y\mathbb{E}[f(X)] - \mathbb{E}[f(X) \mid Y]^2 + 2Y\mathbb{E}[f(X) \mid Y]\right] \tag{119}$$
$$= 2\left(\mathbb{E}\left[Y\mathbb{E}[f(X) \mid Y] - Y\mathbb{E}[f(X)]\right]\right) - \mathbb{E}\left[\mathbb{E}[f(X) \mid Y]^2 + \mathbb{E}[f(X)]^2\right] \tag{120}$$
$$= 2\left(\mathbb{E}\left[Yf(X)\right] - \mathbb{E}[Y]\mathbb{E}[f(X)]\right) - \mathbb{E}\left[\mathbb{E}[f(X) \mid Y]^2 + \mathbb{E}[f(X)]^2\right] \tag{121}$$
$$= 2\mathrm{Cov}(Y, f(X)) - \mathbb{E}\left[\mathbb{E}[f(X) \mid Y]^2\right] + \mathbb{E}[f(X)]^2 \tag{122}$$
$$= 2\mathrm{Cov}(Y, f(X)) - \mathrm{Var}(\mathbb{E}[f(X) \mid Y]) \tag{123}$$
$$\leq 2\alpha \tag{124}$$

Where each step until (124) follows by simply grouping terms and applying linearity of expectation. (124) follows by the multicalibration condition and the fact that the variance of any random variable is nonnegative.

$\square$

**Proof of Lemma C.3**

*Proof.* For any $\pi \in \Pi$, $f \in \mathcal{F}$, assumption (40) gives us $|\text{Cov}(Y, f(X)\pi(X))| \leq \alpha$. We'll expand the LHS to show the result.

$$|\text{Cov}_k(Y, f(X)\pi(X))| \tag{125}$$
$$= |\mathbb{E}_k[\text{Cov}_k(Y, f(X)\pi(X) \mid \pi(X))] + \text{Cov}_k(\mathbb{E}_k[Y \mid \pi(X)], \mathbb{E}_k[f(X)\pi(X) \mid \pi(X)])| \tag{126}$$
$$= |\mathbb{P}_k(\pi(X) = 1)\text{Cov}_k(Y, f(X)\pi(X) \mid \pi(X) = 1)$$
$$\quad + \mathbb{P}_k(\pi(X) = 0)\text{Cov}_k(Y, f(X)\pi(X) \mid \pi(X) = 0)$$
$$\quad + \text{Cov}_k(\mathbb{E}_k[Y \mid \pi(X)], \mathbb{E}_k[f(X)\pi(X) \mid \pi(X)])| \tag{127}$$
$$= |\mathbb{P}(\pi(X) = 1)\text{Cov}_k(Y, f(X) \mid \pi(X) = 1)$$
$$\quad + \text{Cov}_k(\mathbb{E}_k[Y \mid \pi(X)], \mathbb{E}_k[f(X)\pi(X) \mid \pi(X)])| \tag{128}$$

Where (126) is the application of the law of total covariance. Observe now that $\text{Cov}_k(Y, f(X) \mid \pi(X) = 1)$ is exactly what we want to bound. To do so, we now focus on expanding $\text{Cov}_k(\mathbb{E}_k[Y \mid \pi(X)], \mathbb{E}_k[f(X)\pi(X) \mid \pi(X)])$. This is:

$$\mathbb{E}_k[\mathbb{E}_k[Y \mid \pi(X)]\mathbb{E}_k[f(X)\pi(X) \mid \pi(X)]] - \mathbb{E}_k[\mathbb{E}_k[Y \mid \pi(X)]]\mathbb{E}_k[\mathbb{E}_k[f(X)\pi(X) \mid \pi(X)]] \tag{129}$$
$$= \mathbb{P}(\pi(X) = 1)\mathbb{E}_k[Y \mid \pi(X) = 1]\mathbb{E}_k[f(X) \mid \pi(X) = 1]$$
$$\quad - \mathbb{E}_k[Y]\mathbb{P}(\pi(X) = 1)\mathbb{E}_k[f(X) \mid \pi(X) = 1] \tag{130}$$
$$= \mathbb{P}(\pi(X) = 1)\mathbb{E}_k[f(X) \mid \pi(X) = 1] (\mathbb{E}_k[Y \mid \pi(X) = 1] - \mathbb{E}_k[Y]) \tag{131}$$

Because $\pi(\cdot)$ is a binary valued function, we can apply Lemma C.1 to write

$$\mathbb{E}_k[Y \mid \pi(X) = 1] - \mathbb{E}_k[Y] = \frac{\text{Cov}_k(Y, \pi(X))}{\mathbb{P}(\pi(X) = 1)}$$

Plugging in this identity yields:

$$\mathbb{P}(\pi(X) = 1)\mathbb{E}_k[f(X) \mid \pi(X) = 1] (\mathbb{E}_k[Y \mid \pi(X) = 1] - \mathbb{E}_k[Y]) \tag{132}$$
$$= \mathbb{E}_k[f(X) \mid \pi(X) = 1]\text{Cov}_k(Y, \pi(X)) \tag{133}$$

Plugging (133) into (128) yields:

$$|\text{Cov}_k(Y, f(X)\pi(X))| \tag{134}$$
$$= |\mathbb{P}(\pi(X) = 1)\text{Cov}_k(Y, f(X) \mid \pi(X) = 1) + \mathbb{E}_k[f(X) \mid \pi(X) = 1]\text{Cov}_k(Y, \pi(X))| \tag{135}$$
$$= |\mathbb{P}(\pi(X) = 1)\text{Cov}_k(Y, f(X) \mid \pi(X) = 1) - (-\mathbb{E}_k[f(X) \mid \pi(X) = 1]\text{Cov}_k(Y, \pi(X)))|$$
$$\tag{136}$$
$$\geq ||\mathbb{P}(\pi(X) = 1)\text{Cov}_k(Y, f(X) \mid \pi(X) = 1)| - |\mathbb{E}_k[f(X) \mid \pi(X) = 1]\text{Cov}_k(Y, \pi(X))|| \tag{137}$$
$$\geq |\mathbb{P}(\pi(X) = 1)\text{Cov}_k(Y, f(X) \mid \pi(X) = 1)| - |\mathbb{E}_k[f(X) \mid \pi(X) = 1]\text{Cov}_k(Y, \pi(X))| \tag{138}$$

Where (137) is the application of the reverse triangle inequality. Combining the initial assumption that $S_k$ is indistinguishable with respect to $\{f(X)\pi(X) \mid f \in \mathcal{F}, \pi \in \Pi\}$ (40) and (138) yields:

$$|\mathbb{P}(\pi(X) = 1)\text{Cov}_k(Y, f(X) \mid \pi(X) = 1)|$$

$$- |\mathbb{E}_k[f(X) \mid \pi(X) = 1]\mathrm{Cov}_k(Y, \pi(X))| \le \alpha \tag{139}$$

Which further implies:

$$|\mathbb{P}(\pi(X) = 1)\mathrm{Cov}_k(Y, f(X) \mid \pi(X) = 1)| \le \alpha + |\mathbb{E}_k[f(X) \mid \pi(X) = 1]\mathrm{Cov}_k(Y, \pi(X))| \tag{140}$$
$$= \alpha + \mathbb{E}_k[f(X) \mid \pi(X) = 1] |\mathrm{Cov}_k(Y, \pi(X))| \tag{141}$$
$$\le \alpha + |\mathrm{Cov}_k(Y, \pi(X))| \tag{142}$$
$$\le 2\alpha \tag{143}$$

Which finally implies $|\mathrm{Cov}_k(Y, f(X) \mid \pi(X) = 1)| \le \frac{2\alpha}{\mathbb{P}(\pi(X)=1)}$, as desired. (141) and (142) follow from the assumption that $f(X) \in [0, 1]$, and (143) follows from the initial assumption that $S_k$ is $\alpha$-indistinguishable with respect to every $\Pi$ (39). $\qquad\square$

**Proof of Lemma D.3**

*Proof.* Recall that $\tilde{Y}$ is a discrete random variable taking values $0, \frac{\epsilon}{2}, \epsilon, \frac{3\epsilon}{2} \ldots \lfloor \frac{2}{\epsilon} \rfloor \frac{\epsilon}{2}$. We again use $\mathcal{R}(\tilde{Y})$ to denote the range of $\tilde{Y}$. Our analysis below proceeds conditional on the event $\{X \in S\}$, which we suppress for clarity. We can show

$$\left|\mathrm{Cov}(\tilde{Y}, f(X))\right| = \left|\mathbb{E}[\tilde{Y}f(X)] - \mathbb{E}[\tilde{Y}]\mathbb{E}[f(X)]\right| \tag{144}$$

$$= \left|\mathbb{E}[\tilde{Y}f(X)] - \mathbb{E}[f(X)] \sum_{\tilde{y} \in \mathcal{R}(\tilde{Y})} \tilde{y}\mathbb{P}(\tilde{Y} = \tilde{y})\right| \tag{145}$$

$$= \left|\mathbb{E}[\tilde{Y}\mathbb{E}[f(X) \mid \tilde{Y}]] - \mathbb{E}[f(X)] \sum_{\tilde{y} \in \mathcal{R}(\tilde{Y})} \tilde{y}\mathbb{P}(\tilde{Y} = \tilde{y})\right| \tag{146}$$

$$= \left|\sum_{\tilde{y} \in \mathcal{R}(\tilde{Y})} \tilde{y}\mathbb{P}(\tilde{Y} = \tilde{y})\mathbb{E}[f(X) \mid \tilde{Y} = \tilde{y}] - \mathbb{E}[f(X)] \sum_{\tilde{y} \in \mathcal{R}(\tilde{Y})} \tilde{y}\mathbb{P}(\tilde{Y} = \tilde{y})\right| \tag{147}$$

$$= \left|\sum_{\tilde{y} \in \mathcal{R}(\tilde{Y})} \tilde{y}\mathbb{P}(\tilde{Y} = \tilde{y}) \left(\mathbb{E}[f(X) \mid \tilde{Y} = \tilde{y}] - \mathbb{E}[f(X)]\right)\right| \tag{148}$$

$$= \sum_{\tilde{y} \in \mathcal{R}(\tilde{Y})} \tilde{y}\mathbb{P}(\tilde{Y} = \tilde{y}) \left|\left(\mathbb{E}[f(X) \mid \tilde{Y} = \tilde{y}] - \mathbb{E}[f(X)]\right)\right| \tag{149}$$

$$= \sum_{\tilde{y} \in \mathcal{R}(\tilde{Y})} \tilde{y} \left|\mathrm{Cov}(1(\tilde{Y} = \tilde{y}), f(X))\right| \tag{150}$$

$$\le \sum_{\tilde{y} \in \mathcal{R}(\tilde{Y})} \tilde{y}\alpha \tag{151}$$

$$\le \sum_{\tilde{y} \in \mathcal{R}(\tilde{Y})} \alpha \tag{152}$$

$$\le \left\lceil \frac{2}{\epsilon} \right\rceil \alpha \tag{153}$$

Where (149) makes use of the fact that $\tilde{y} \ge 0$, (150) makes use of the identity $\left|\mathrm{Cov}(1(\tilde{Y} = \tilde{y}), f(X))\right| = \mathbb{P}(\tilde{Y} = \tilde{y}) \left|\left(\mathbb{E}[f(X) \mid \tilde{Y} = \tilde{y}] - \mathbb{E}[f(X)]\right)\right|$ (this is a straightforward analogue of Lemma C.1), (151) applies assumption (94), and (152) makes use of the fact that $\tilde{y} \le 1$.

$\qquad\square$

## F  Finding multicalibrated partitions: chest X-ray diagnosis

In this section we provide additional details related to the chest X-ray diagnosis task studied in Section 5. As discussed in Section 5, the relevant class $\mathcal{F}$ is the set of 8 predictive models studied in [59] for diagnosing various pathologies using chest X-ray imaging data. A key insight for learning a multicalibrated partition with respect to this class is that, because the class of models is finite, we can apply Lemma B.1 to discover indistinguishable subsets without training new models or even accessing the training data. Instead, we can think of the relevant input space as $\{0,1\}^8$: the 8-dimensional vector containing the classifications output by the 8 models in $\mathcal{F}$ for a given X-ray. Then, per Lemma Lemma C.3 and Corollary A.3, any subset of X-rays for which the Chebyshev distance (i.e., the maximum coordinatewise difference) in this 8-dimensional space is bounded must be approximately indistinguishable. Thus, to find approximately indistinguishable subsets, we simply apply an off-the-shelf clustering algorithm to minimize the intracluster Chebyshev distance. Code and instructions to replicate this procedure are available at `https://github.com/ralur/heap-repl`.

## G  Additional experimental results: chest X-ray diagnosis

In this section we provide results which are analagous to those presented in Section 5 for the four additional pathologies studied in [59]. For each pathology we first present a figure comparing the accuracy of the benchmark radiologists to that of the eight leaderboard algorithms, as in Figure 2 for atelectasis. We then present a figure which plots the conditional performance of each radiologist within a pair of indistinguishable subsets, as in Figure 3.

Results for diagnosing a pleural effusion are presented in Figure 5 and Figure 6. Results for diagnosing cardiomegaly are presented in Figure 7 and Figure 8. Results for diagnosing consolidation are presented in Figure 9 and Figure 10. Finally, results for diagnosing edema are presented in Figure 11 and Figure 12.

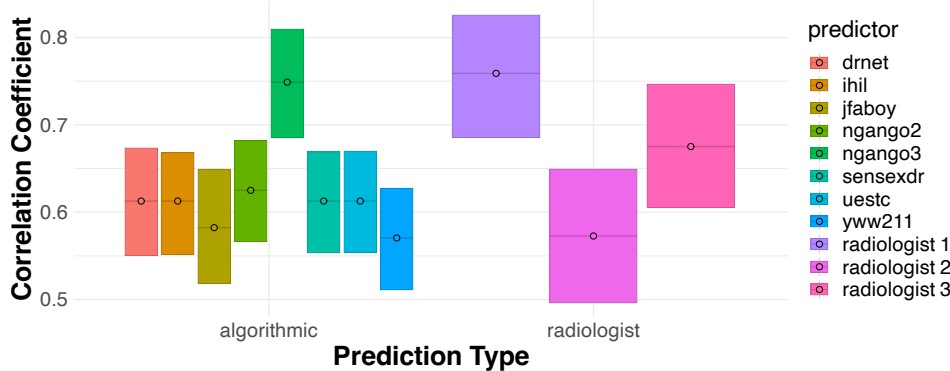

Figure 5: The relative performance of radiologists and predictive algorithms for detecting a pleural effusion. Each bar plots the Matthews Correlation Coefficient between the corresponding prediction and the ground truth label. Point estimates are reported with $95\%$ bootstrap confidence intervals.

## H  Additional experimental details: prediction from visual features

In this section we provide additional details related to the escape the room task studied in Section 5. Our work focuses on study 2 in [62]; study 1 analyzes the same task with only two treatment arms. As discussed in Section 5, the dimension of the feature space is 33, and encodes the mean/median/standard deviation (for numerical features) or one-hot encoding (for categorical features) of the following: the location in which the puzzle was attempted (Boston, Arizona, NYC etc.), the type of escape the room puzzle (theater, office, home etc.), the number of people in the photo, summary demographic information (age, gender, race, racial diversity), whether participants are smiling, and whether (and what type) of glasses they are wearing. This is exactly the set of features considered in [62]; for additional detail on the data collection process we refer to their work.

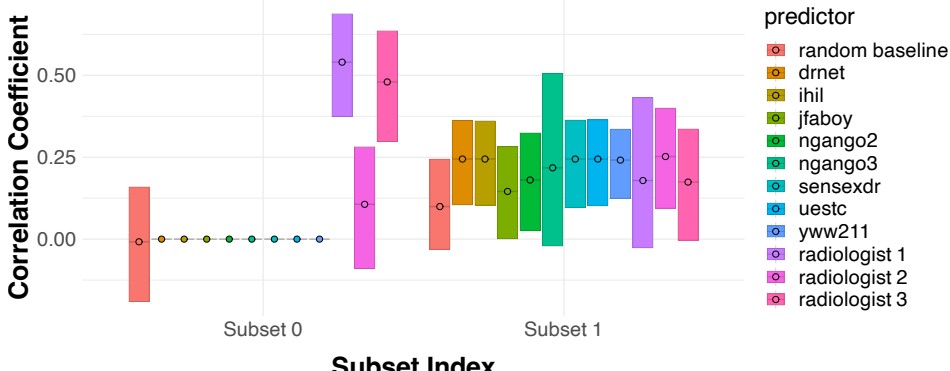

Figure 6: The conditional performance of radiologists and predictive algorithms for detecting a pleural effusion within two approximately indistinguishable subsets. A random permutation of the true labels is included as a baseline. The confidence intervals for the algorithmic predictors are not strictly valid (the subsets are chosen conditional on the predictions themselves), but are included for reference against radiologist performance. All else is as in Figure 5.

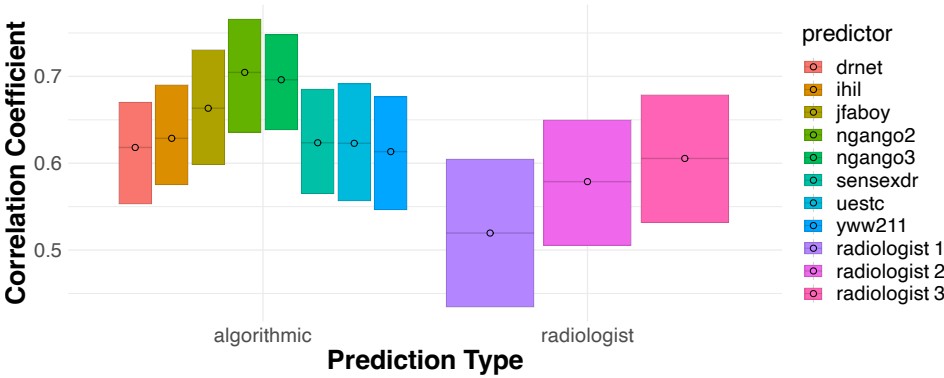

Figure 7: The relative performance of radiologists and predictive algorithms for detecting cardiomegaly. Each bar plots the Matthews Correlation Coefficient between the corresponding prediction and the ground truth label. Point estimates are reported with $95\%$ bootstrap confidence intervals.

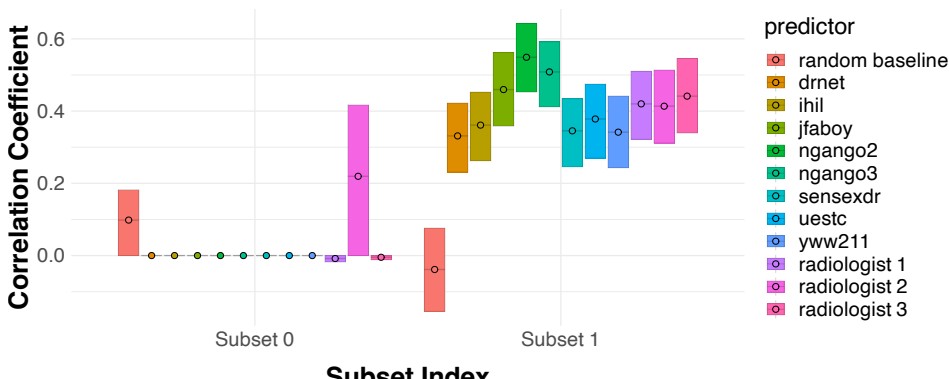

Figure 8: The conditional performance of radiologists and predictive algorithms for detecting cardiomegaly within two approximately indistinguishable subsets. A random permutation of the true labels is included as a baseline. The confidence intervals for the algorithmic predictors are not strictly valid (the subsets are chosen conditional on the predictions themselves), but are included for reference against radiologist performance. All else is as in Figure 7.

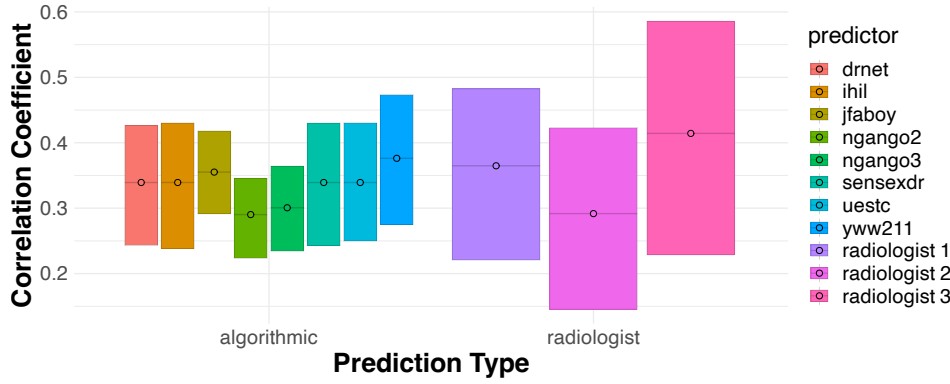

Figure 9: The relative performance of radiologists and predictive algorithms for detecting consolidation. Each bar plots the Matthews Correlation Coefficient between the corresponding prediction and the ground truth label. Point estimates are reported with 95% bootstrap confidence intervals.

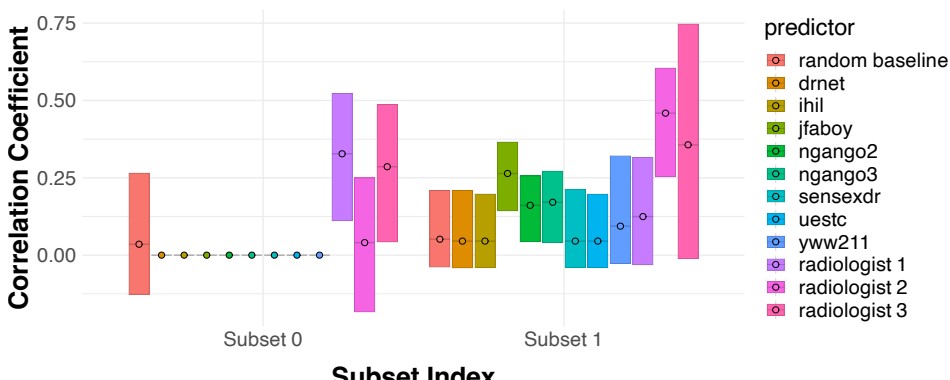

Figure 10: The conditional performance of radiologists and predictive algorithms for detecting consolidation within two approximately indistinguishable subsets. A random permutation of the true labels is included as a baseline. The confidence intervals for the algorithmic predictors are not strictly valid (the subsets are chosen conditional on the predictions themselves), but are included for reference against radiologist performance. All else is as in Figure 9.

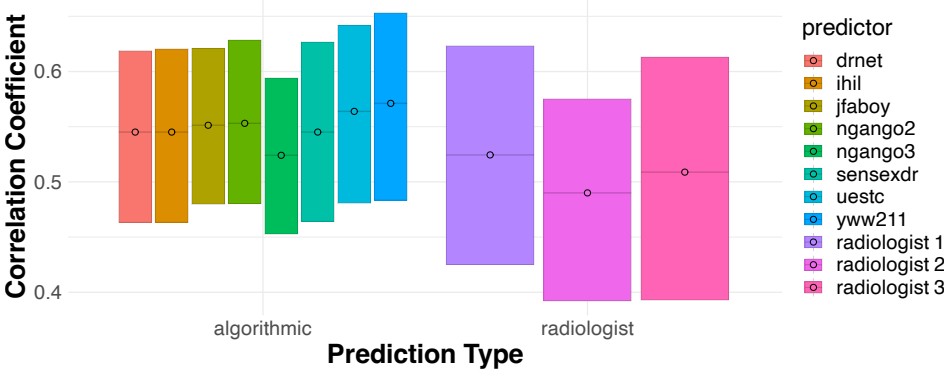

Figure 11: The relative performance of radiologists and predictive algorithms for detecting edema. Each bar plots the Matthews Correlation Coefficient between the corresponding prediction and the ground truth label. Point estimates are reported with 95% bootstrap confidence intervals.

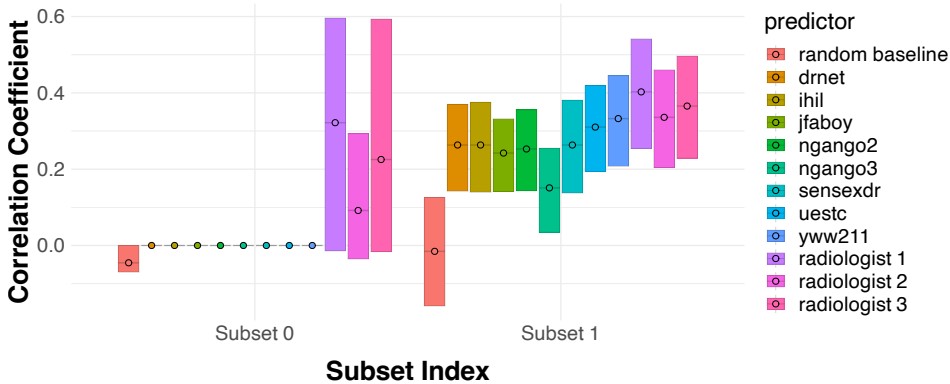

Figure 12: The conditional performance of radiologists and predictive algorithms for detecting edema within two approximately indistinguishable subsets. A random permutation of the true labels is included as a baseline. The confidence intervals for the algorithmic predictors are not strictly valid (the subsets are chosen conditional on the predictions themselves), but are included for reference against radiologist performance. All else is as in Figure 11.

To learn a partition of the input space, we apply the boosting algorithm proposed in [27]. We discuss this algorithm and its connection to multicalibration in Appendix B.

# I  Additional experimental results: prediction from visual features

In this section we present additional experimental results for the visual prediction task studied in [62].

**Humans fail to outperform algorithms.** As in the X-ray diagnosis task in Section 5, we first directly compare the performance of human subjects to that of the five off-the-shelf learning algorithms studied in [62]. We again use the Matthew's Correlation Coefficient (MCC) as a measure of binary classification accuracy [12]. Our results confirm one of the basic findings in [62], which is that humans fail to outperform the best algorithmic predictors. We present these results in Figure 13.

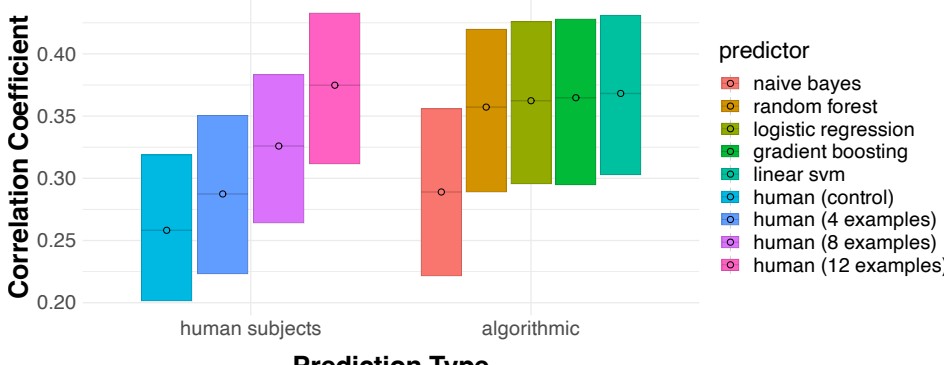

Figure 13: Comparing the accuracy of human subjects' predictions to those made by off-the-shelf learning algorithms across four treatment conditions. Subjects in the control condition are given no training, while subjects in each of the three remaining conditions are presented with a small number of labeled examples before beginning the task. Each bar plots the Matthews correlation coefficient between the corresponding prediction and the true binary outcome; point estimates are reported with 95% bootstrap confidence intervals.

Although these results indicate that humans fail to outperform algorithms on average in this visual prediction task, we now apply the results of Section 4 to investigate whether humans subjects can refine algorithmic predictions on *specific* instances.

**Resolving indistinguishability via human judgment.** As in Section 5, we first form a partition of the set of input images which is multicalibrated with respect to the five predictors considered in Figure 13. As indicated by Lemma B.1 and Corollary B.2, we do this by partitioning the space of observations to minimize the variance of each of the five predictors within each subset.[11] Because the outcome is binary, it is natural to partition the space of images into two clusters. We now examine the conditional correlation between each prediction and the true binary outcome within each of these subsets, which we plot in Figure 14.

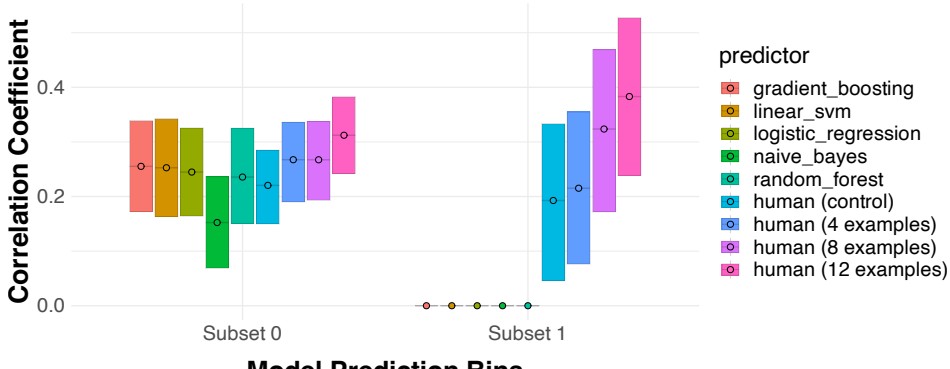

Figure 14: The conditional performance of human and algorithmic predictions within two approximately indistinguishable subsets. Subset 1 ($n = 189$) is the set in which all five predictors predict a positive label; Subset 0 ($n = 671$) contains the remaining observations. All else is as in Figure 13. The confidence intervals for the algorithmic predictors are not strictly valid (the subsets are chosen conditional on the predictions themselves), but are included for reference against human performance.

As we can see, the human subjects' predictions perform comparably to the algorithms within subset 0, but add substantial additional signal when all five models predict a positive label (subset 1). Thus, although the human subjects fail to outperform the algorithmic predictors on average (Figure 13), there is substantial heterogeneity in their relative performance that can be *identified ex-ante* by partitioning the observations into two approximately indistinguishable subsets. In particular, as in the X-ray classification task studied in Section 5, we find that human subjects can identify negative instances which are incorrectly classified as positive by all five algorithmic predictors.

---

[11]We describe this procedure in Appendix F.

