# OpenReview forum: "Human Expertise in Algorithmic Prediction"
_NeurIPS.cc/2024/Conference — NeurIPS 2024 oral_

### Official Review · Reviewer_XFzN · 2024-06-25

**Soundness:** 4
**Presentation:** 4
**Contribution:** 4
**Rating:** 8
**Confidence:** 4

**Summary:**

This paper introduces a new framework into algorithmic predictions. The paper asks and answers the question "how can we incorporate human input into the prediction algorithm, which may not even be captured in the training data"? The authors develop a method that first runs the predictor, and then runs a second predictor using the human input. The authors show that even a simple instantiation of their method can outperform existing predictors. They use the X-ray classification task as experimental datasets.

**Strengths:**

The paper is written very clearly, and offers a novel method to incorporate human input into algorithmic prediction. Both theoretical derivations and experiment results are sound. The contributions of this paper is significant, and I believe this paper deserves to be accepted in its current form.

**Weaknesses:**

The paper would be even more satisfying if the method is presented as a framework rather than a specific instantiation. In addition, it would be great if the authors can discuss potential ways to improve on the method they propose, and what these methods mean in the broader context of incorporating human feedback into algorithmic predictions. Nevertheless, these small weaknesses does not diminish the significance and novelty of this paper.

**Questions:**

My main comment is that the authors should comment more about the future work and implications of this method. Furthermore, I would be interested to hear what the authors think about a related paper [1], and how these papers might be related.

[1] DEFINING EXPERTISE: APPLICATIONS TO TREATMENT EFFECT ESTIMATION (https://arxiv.org/pdf/2403.00694)

**Limitations:**

The authors have addressed the limitations in the conclusion section

---

> ### Author Rebuttal · Authors · 2024-08-06
>
> **In response to:** *The paper would be even more satisfying if the method is presented as a framework rather than a specific instantiation...* And: *My main comment is that the authors should comment more about the future work and implications of this method.*
>
> Thank you for your feedback, we agree that we could devote more space to discussing the generality of our approach and avenues for possible future work. We provide our thoughts on this point below, and will plan to update our manuscript to better communicate the scope of indistinguishability beyond minimizing mean squared error (this is closely related to reviewer fseb's question about modeling decision makers which richer preferences; for convenience, we copy our response under both reviewers comments).
>
> One way of interpreting algorithmic indistinguishability is that the algorithm provides the decision maker with a partial ordering over instances, where the ordering is defined with respect to a single outcome of interest $Y$. In particular, the algorithm can assert that instances in one indistinguishable subset $S_1$ have a larger average value of $Y$ than another indistinguishable subset $S_2$ --- so the algorithm implicitly ranks each $x \in S_1$ higher than each $x \in S_2$ --- but it has no way of ordering instances *within* each indistinguishable subset. Theorem 4.1 and Corollary 4.2 focus on settings where the objective of interest is to minimize mean squared error, and show how additional information provided by the expert can be used in service of this objective. However, this is far from the only possibility: for example, a decision maker might use predictions to inform a selection rule which seeks to balance both maximizing the mean value of $Y$ within the pool selected while also ensuring some measure of fairness or diversity (e.g., a university choosing which students to accept, where $Y$ is some measure of academic performance). In this setting, a very natural application of our framework would be to present the decision maker with a set of inputs which are algorithmically indistinguishable with respect to $Y$, and allow the decision maker to then choose from this pool to maximize their chosen fairness metric (e.g., by choosing a set of candidates with diverse interests from within a pool which cannot be distinguished on the basis of predicted academic performance). Similarly, a decision maker whose utility function includes some measure of risk aversion may select a pool of candidates from within an indistinguishable subset to minimize e.g., the *variance* of $Y$ among those selected. In both cases, indistinguishability provides a principled basis for imposing "secondary" preferences in decision making, as the decision maker can reasonably assert that they otherwise lack information to distinguish instances on the basis of (the expected value of) $Y$ alone.
>
> Finally, we note that in both of these examples, we did not assume that the decision maker's utility can be linearly decomposed across inputs. This is in contrast to mean squared error, which can be decomposed as a (normalized) *sum* of prediction errors accross inputs. For example, a measure of fairness might depend on the composition of the entire group selected; it may not always make sense to ask whether the selection of a single individual in isolation is "fair". Similarly, a measure of risk might also depend on the composition of the entire group which is selected; perhaps the decision maker wants to select a set of inputs whose outcomes are minimally correlated (e.g., choosing a portfolio of stocks), and thus their utility is again necessarily a set-valued function.  Thus, our framework is not restricted to minimizing mean squared error or simple variants thereof; instead, it provides a substantially more general basis for decision making under uncertainty. Modeling a decision maker with richer preferences is a fascinating direction, and would be happy to address any followup questions or comments during the upcoming discussion period.
>
>
> **In response to:** *Furthermore, I would be interested to hear what the authors think about a related paper [1], and how these papers might be related. https://arxiv.org/pdf/2403.00694*
>
> Thank you for the pointer to this work. This is a fascinating and related topic, which characterizes the value of human expertise as an inductive bias for guiding model selection. In particular, the authors argue that expert decisions regarding treatment allocation can be highly predictive of the true treatment effects, and that this fact can be used to inform the model selection process for treatment effect estimation. Importantly however --- and as is typical in treatment effect estimation problems --- [1] assumes *no hidden confounding* (see Section 2), which rules out the possibility that experts leverage information which is correlated with potential outcomes but unavailable to the algorithm.
>
> This perspective is complementary to our work, which is instead focused on the possibility that experts *do* have information which is unavailable to the algorithm, and studies how to incorporate this information at "inference" or "test" time. Thus, even given infinite data to learn the best possible (e.g., Bayes optimal) model of $Y$ given $X$, our method might incorporate expert feedback at test time to improve predictions. In contrast, [1] uses expert decisions at training time to more efficiently learn a model under sample size constraints, but do not consider incorporating expertise at test time because it is assumed that experts do not provide signal which the algorithm could not (eventually, given sufficient training data, and under the overlap and unconfoundedness assumptions) learn on its own.
>
> We thank you for pointing us to this work, and we will include a citation in our manuscript. We'd be happy to discuss this point further and answer any followup questions during the upcoming discussion period.

---

> > ### Comment · Reviewer_XFzN · 2024-08-11
> > **I recommend acceptance**
> >
> > Many thanks for the authors' detailed response. I am happy to see that all reviewers unanimously recommended acceptance. Therefore, I am happy to accept the paper, and nominate the paper for awards if the AC agrees.

---

### Official Review · Reviewer_WEMo · 2024-07-07

**Soundness:** 3
**Presentation:** 4
**Contribution:** 3
**Rating:** 7
**Confidence:** 4

**Summary:**

The paper proposes a framework to incorporate human expert knowledge in algorithmic predictions. Under this framework, the authors introduce a meta-algorithm that uses a training dataset including human expert predictions together with a multi calibrated partition of the data; a partition of the dataset into bins, where each bin contains data that are indistinguishable to the predictive model. Using the data of each bin the meta-algorithm trains a regression algorithm to predict the true label from the human expert prediction. In this way, the authors aim to leverage the human expertise, that may be more accurate than the predictive algorithm on specific instances, to achieve complimentary—to achieve higher predictive accuracy through human AI collaboration than the performance of a human expert or AI in isolation.

**Strengths:**

The paper suggests an elegant method to improve algorithmic predictions in light of human expertise, that could have significant applications such as the medical domain, where the additional information of human experts may lead them to more accurate predictions on certain instances compared ot predictive models.

The paper is very well and clearly written, nicely motivated and follows a clear structure. There is a thorough and comprehensive discussion on related work as well as a comprehensive and clearly presented experimental evaluation.

**Weaknesses:**

Since the theoretical results of section 6 complement the ones of section 4, it would be perhaps more natural to follow them, rather than placing them after the experimental evaluation, which appears a bit odd.

**Questions:**

N/A

**Limitations:**

The authors adequately discuss the limitations of their work.

---

> ### Author Rebuttal · Authors · 2024-08-06
>
> **In response to:** *Since the theoretical results of section 6 complement the ones of section 4, it would be perhaps more natural to follow them, rather than placing them after the experimental evaluation, which appears a bit odd.*
>
> Thank you for your feedback --- we agree that this portion of the manuscript could flow a bit better. Section 4 and 5 are intended to be complementary, as both focus on a particular application of algorithmic indistinguishability. In contrast, section 6 presents results on a qualitatively different application of indistinguishability, which is intended to highlight the generality of our framework and suggest open directions for future work. We will update both the exposition to section 6 (see response to reviewer jnGw) and more directly highlight the flexibility of our approach (see responses to reviewers fseb and XFzN), which we expect will address this concern as well. We are however certainly open to other feedback on this point.

---

> > ### Comment · Reviewer_WEMo · 2024-08-09
> >
> > I would like to thank the authors for their reply. The suggested changes by the authors address my point and should be done to improve the flow.

---

### Official Review · Reviewer_jnGw · 2024-07-13

**Soundness:** 3
**Presentation:** 3
**Contribution:** 3
**Rating:** 7
**Confidence:** 3

**Summary:**

The paper first presents some theory for the modelling of how to identify when human judgements may offer a better diagnosis - through access to additional information - than machine predictions, despite the latter typically being more accurate. This is followed by exploring how to integrate the human input with the algorithmic (model) input. Subsequently, the authors present some focussed experimental results using chest x-ray interpretation that support their proposition.

**Strengths:**

Originality: carefully drawn comparison with the literature, situates and differentiates the contribution.

Quality + Clarity (addressed together):

Clear abstract and intro with well-defined contributions. Content offers a reasonable balance between technical and intuitive. Recognition of the value of the human contribution and seeking to integrate it in decision making.

The later mathematical results (section 4) have effective accompanying interpretations (see complementary point in weaknesses).

Effective, selective presentation of results: choosing one and going into detail, while two other cases in the appendices support the same observation, rather than trying to squeeze them all into the paper body. Same applies to results in section 5.2.

Significance: provides a sound framework for a particular, amenable class of collaboration problems that allows for the proper incorporation of human prediction where machine prediction could fall short.

**Weaknesses:**

Clarity: Indistinguishability and multicalibration are critical elements to the contribution; it would be helpful if the interpretation of their definitions (3.1, 3.2) went into a bit more detail for accessibility.

This reader is not succeeding in following the argument about robustness (section 6).

**Questions:**

Q1. The case studies are retrospective so both machine and human outcomes are available to use in the analysis. How would the approach work in a live situation?

**Limitations:**

Section 7 provides some properly reflective critique on scope and applicability.

---

> ### Author Rebuttal · Authors · 2024-08-06
>
> **In response to:** *...it would be helpful if the interpretation of...definitions (3.1, 3.2) went into a bit more detail for accessibility.*
>
> Thank you for your feedback. We agree that these definitions could use a bit more exposition; we will add more background and provide a concrete interpretation in the context of diagnosing atelectasis (which is described earlier in the introduction) to make them more accessible.
>
> **In response to:** *This reader is not succeeding in following the argument about robustness (section 6).*
>
> We agree that this portion of the manuscript could be more accessible. We will add some additional exposition to section 6 and further clarify the intended application of these results. To motivate this section, it is helpful to consider the following stylized example:
>
> Suppose we are interested in designing an algorithmic risk score for a large hospital system. As discussed in section 6, one important consideration is that doctors at this hospital system may only selectively comply with the algorithm. Concretely, suppose there is one doctor, doctor A, who generally complies with the algorithm's recommendations except on patients who have high blood pressure; the doctor believes (correctly or not) that the algorithm underweights high blood pressure as a risk factor, and simply uses their judgment to make decisions for this subset of patients. A second doctor, doctor B, similarly complies with the algorithm on all patients under the age of 65, but ignores its recommendations for patients 65 and older.
>
> What does an optimal algorithm look like? For doctor A, we would like to select the algorithm which minimizes error on patients who do not have high blood pressure, as these are the patients for whom the doctor actually uses the algorithm's recommendations. Similarly, for doctor B, we would like to select the algorithm which minimizes error on patients who are under 65. The key thing to note is that there is no guarantee that these are the same algorithm: if we were to run empirical risk minimization over the first population of patients, we might get a different predictor than if we ran empirical risk minimization over the second population. This is of course not just a finite sample problem; it is also possible that, given any restricted model class (e.g., all linear predictors), the optimal predictor for one subpopulation may not be optimal for a different subpopulation.
>
> For both practical and ethical reasons, we cannot provide individualized risk scores for every physician; we must provide a single risk predictor for the entire hospital system. Our first result (Lemma A.4) shows that, without further assumptions, this is an intractable problem. If there are arbitrarily many physicians who may choose to comply in arbitrary ways, then we need a predictor which is simultaneously optimal for every possible patient subpopulation. The only predictor which satisfies this criterion is the Bayes optimal predictor, which is infeasible to learn in a finite data regime.
>
> However, it is perhaps more likely that physicians decide whether or not to defer to the algorithm using relatively simple heuristics. If we believe we can model these heuristics as a "simple" function of observable patient characteristics --- e.g., that all compliance patterns can be expressed as a shallow decision tree, even if particular compliance behavior varies across physicians --- then perhaps we can leverage this structure to design a single optimal predictor. Theorem 6.1 shows that indeed this is possible: if a predictor is multicalibrated over the appropriate class of risk scores and compliance patterns, then, for every physician, on the subpopulation of patients that the physician defers to the algorithm, the performance of the multicalibrated predictor is competitive with that of any other predictor in the class ${\cal F}$. Thus, by leveraging known or assumed structure in user compliance patterns, we can design predictors which are "robust" to those compliance patterns.
>
> We hope that helps clarify the motivation and intended application of the results in section 6. As mentioned above, we will update our manuscript to make this section more accessible, and would be happy to answer any follow up questions during the upcoming discussion period.
>
> **In response to:** *The case studies are retrospective so both machine and human outcomes are available to use in the analysis. How would the approach work in a live situation?*
>
> This is an excellent question, and one which we will address in more detail (particularly in the discussion section). On a forward-looking basis, perhaps the most natural application of our framework is to proactively solicit expert feedback on *only* those inputs where it appears to provide additional information. For example, in section 4, we observe that a nonzero coefficient $\beta_k^*$ indicates that an expert provides additional signal within subset $S_k$, whereas a coefficient of $0$ indicates the opposite. Thus, although we assume that we have retrospective expert predictions for every instance, we could prospectively focus expert attention on only the subset of instances where experts add value and simply defer the remainder to an algorithm.
>
> More generally, choices for both *whether* to solicit expert feedback and *how* to incorporate that feedback into forward-looking decisions is a substantial topic in its own right. For example, translating predictions to decisions, and deciding whether to incorporate expert feedback to do so, may require a rich model of decision makers' preferences or utility functions. We discuss these at length in our response to reviewers fseb and XFzN. Furthermore, although we focus on the standard "batch" supervised learning setting to highlight our main contributions, we view the extension to an online learning context as a promising avenue for future work, and will include it in our discussion of open problems in section 7.

---

> > ### Comment · Reviewer_jnGw · 2024-08-13
> >
> > Thanks for the detailed and helpful response both to me and the other reviewers.

---

> > ### Public Comment · ~Feiyu_Zhu2 · 2024-12-19
> >
> > Hello, thank you for your interesting paper, but as for your example of Doctor A and Doctor B, I think it is the reverse causality. In your example, Doctor A, except for patients with hypertension, usually follows the advice of the algorithm. This situation is often because the algorithm has more errors in the judgment of patients with hypertension, which is not as accurate as the doctor's own judgment. At other times, the judgment is accurate, and the doctor does not need to invest too much effort and judgment. You deduce from this that the optimal algorithm is minimizes error on patients who do not have high blood pressure. I think this judgment is irresponsible. It is difficult for me to understand how the example you give contributes to the explanation of Section VI

---

> > > ### Public Comment · ~Rohan_Alur1 · 2024-12-19
> > >
> > > Thank you for your comment. The example is correct as written — the physician only uses the algorithm’s recommendations on a subset of the distribution, and our goal is to minimize error on this subset. Our results are agnostic as to why this is the case; in particular, we do not interrogate whether the physician is “correct” in their judgment. Furthermore, these kinds of restrictions can arise for other reasons — for example, it may be that a certain physician (or hospital) is not equipped to see patients with a certain condition, in which case a similar restriction arises. We take this conditioning event as given, and seek to minimize error over the corresponding conditional distribution.

---

### Official Review · Reviewer_fseb · 2024-07-18

**Soundness:** 4
**Presentation:** 4
**Contribution:** 4
**Rating:** 8
**Confidence:** 3

**Summary:**

This paper introduces a framework for joint human-AI prediction, where human experts can augment AI predictions in particular ex ante identifiable subsets.

**Strengths:**

This paper makes a lot of interesting contributions. First, its scope is broad and important: it tackles the question of how and whether human judgment can improve the predictions of any learning algorithm. That is and will remain to be a very important question in our time. It contributes a very interesting framework, rooted in algorithmic indistinguishability and multicalibration, to find subsets in which no algorithm in a user-specified class has predictive power (because they are algorithmically indistinguishable) but human experts do (because they might have more access to the instances, such as doctors examining patients). It demonstrates that using this framework, we can find subsets of instances where human experts can outperform algorithms, and thus the combination of the two can outperform either alone. It applies this to an important medical problem and in another domain of making predictions from photos of people. It even extends the framework to apply to a setting with noncompliance. The community stands to learn a lot from this paper.

**Weaknesses:**

As the authors mention, the framework is dependent on minimizing mean squared error only.

**Questions:**

How might you model deicision makers with richer preferences than mean squared error?

**Limitations:**

Yes.

---

> ### Author Rebuttal · Authors · 2024-08-06
>
> **In response to:** *How might you model decision makers with richer preferences than mean squared error?*
>
> Thank you for your feedback, we agree that we should address this possibility in more detail (particularly in the final discussion section). We provide our thoughts below, and will plan to update our manuscript to better communicate the scope of indistinguishability beyond minimizing mean squared error (this is closely related to reviewer XFzN's question about the generality of our framework; for convenience, we copy our response under both reviewers comments).
>
> One way of interpreting algorithmic indistinguishability is that the algorithm provides the decision maker with a partial ordering over instances, where the ordering is defined with respect to a single outcome of interest $Y$. In particular, the algorithm can assert that instances in one indistinguishable subset $S_1$ have a larger average value of $Y$ than another indistinguishable subset $S_2$ --- so the algorithm implicitly ranks each $x \in S_1$ higher than each $x \in S_2$ --- but it has no way of ordering instances *within* each indistinguishable subset. Theorem 4.1 and Corollary 4.2 focus on settings where the objective of interest is to minimize mean squared error, and show how additional information provided by the expert can be used in service of this objective. However, this is far from the only possibility: for example, a decision maker might use predictions to inform a selection rule which seeks to balance both maximizing the mean value of $Y$ within the pool selected while also ensuring some measure of fairness or diversity (e.g., a university choosing which students to accept, where $Y$ is some measure of academic performance). In this setting, a very natural application of our framework would be to present the decision maker with a set of inputs which are algorithmically indistinguishable with respect to $Y$, and allow the decision maker to then choose from this pool to maximize their chosen fairness metric (e.g., by choosing a set of candidates with diverse interests from within a pool which cannot be distinguished on the basis of predicted academic performance). Similarly, a decision maker whose utility function includes some measure of risk aversion may select a pool of candidates from within an indistinguishable subset to minimize e.g., the *variance* of $Y$ among those selected. In both cases, indistinguishability provides a principled basis for imposing "secondary" preferences in decision making, as the decision maker can reasonably assert that they otherwise lack information to distinguish instances on the basis of (the expected value of) $Y$ alone.
>
> Finally, we note that in both of these examples, we did not assume that the decision maker's utility can be linearly decomposed across inputs. This is in contrast to mean squared error, which can be decomposed as a (normalized) *sum* of prediction errors accross inputs. For example, a measure of fairness might depend on the composition of the entire group selected; it may not always make sense to ask whether the selection of a single individual in isolation is "fair". Similarly, a measure of risk might also depend on the composition of the entire group which is selected; perhaps the decision maker wants to select a set of inputs whose outcomes are minimally correlated (e.g., choosing a portfolio of stocks), and thus their utility is again necessarily a set-valued function.  Thus, our framework is not restricted to minimizing mean squared error or simple variants thereof; instead, it provides a substantially more general basis for decision making under uncertainty. Modeling a decision maker with richer preferences is a fascinating direction, and would be happy to address any followup questions or comments during the upcoming discussion period.

---

### Author Rebuttal · Authors · 2024-08-06

We are grateful to all four reviewers for their thoughtful and constructive feedback. Below we describe how we intend to incorporate this feedback into our manuscript and include responses to specific reviewer questions and concerns.

---

### Decision · Program_Chairs · 2024-09-25

**Decision:**

Accept (oral)

**Comment:**

The paper tackles an important and longstanding problem: when and how human judgment can improve algorithmic predictions. It identifies a key technical construct: multicalibration subsets where algorithms' predictions will be indistinguishable, but humans can still make distinctions. It uses this insight to create a meta-algorithm to improve the algorithm's predictions. Reviewers appreciated the broad and fundamental insight as well as experiments on three diverse datasets. The paper is also well-written and accessible. Thank you for submitting this work to NeurIPS and I'm glad to recommend Accept.